# Molecular details of ruthenium red pore block in TRPV channels

Ruth A Pumroy[1,5], José J De Jesús-Pérez[1,5], Anna D Protopopova[1,5], Julia A Rocereta[1], Edwin C Fluck[1], Tabea Fricke [2], Bo-Hyun Lee[3,4], Tibor Rohacs [4], Andreas Leffler [2] & Vera Moiseenkova-Bell [1✉]

## Abstract

**Transient receptor potential vanilloid (TRPV) channels play a critical role in calcium homeostasis, pain sensation, immunological response, and cancer progression. TRPV channels are blocked by ruthenium red (RR), a universal pore blocker for a wide array of cation channels. Here we use cryo-electron microscopy to reveal the molecular details of RR block in TRPV2 and TRPV5, members of the two TRPV subfamilies. In TRPV2 activated by 2-aminoethoxydiphenyl borate, RR is tightly coordinated in the open selectivity filter, blocking ion flow and preventing channel inactivation. In TRPV5 activated by phosphatidylinositol 4,5-bisphosphate, RR blocks the selectivity filter and closes the lower gate through an interaction with polar residues in the pore vestibule. Together, our results provide a detailed understanding of TRPV subfamily pore block, the dynamic nature of the selectivity filter and allosteric communication between the selectivity filter and lower gate.**

**Keywords** Channel Activation and Blocking; Cryo-Electron Microscopy; Pore Blocker; Ruthenium Red; TRPV Channels
**Subject Categories** Membranes & Trafficking; Structural Biology

## Introduction

Transient receptor potential vanilloid (TRPV) channels are a group of six polymodal cation channels subdivided into two subgroups by their sequence identity and functional properties (Montell, 2005). TRPV1–TRPV4 comprise the first subgroup of non-selective $Ca^{2+}$ channels, TRPV5 and TRPV6 constitute the second subgroup of $Ca^{2+}$ selective channels (den Dekker et al, 2003; Pumroy et al, 2020). A key feature responsible for the difference in their cation selectivity is the structure of the selectivity filter (SF). Biochemical and structural data show that TRPV1-TRPV4 have a flexible and adaptable SF (Deng et al, 2018; Jara-Oseguera et al, 2019; Pumroy et al, 2022, 2019; Zhang et al, 2021; Zubcevic et al, 2018a, 2018b), which passes ions even when the

channels are closed and can allow for the passage of large organic cations like Yo-Pro1 and doxorubicin through the opened channel (Meyers et al, 2003; Nabissi et al, 2013). Conversely, TRPV5 and TRPV6 have an extremely rigid SF characterized by a proposed knock-off mechanism for cation permeation (Dang et al, 2019; Hughes et al, 2018a; McGoldrick et al, 2018; Sakipov et al, 2018; Saotome et al, 2016). In this mechanism, the SF can host multiple ions simultaneously, then when a new $Ca^{2+}$ ion is introduced into the SF, it displaces or "knocks off" the originally positioned $Ca^{2+}$ ion from the pore, making room for the incoming one (Lopin et al, 2010; Tang et al, 2014). Yet despite this difference, both subgroups can be blocked by the same set of cationic pore blockers, including ruthenium red (RR) (Caterina et al, 1999; Dray et al, 1990; Hoenderop et al, 2001; Nilius et al, 2001a; Peier et al, 2002; Strotmann et al, 2000).

RR is an inorganic dye originally used in histology to increase contrast of negatively charged tissues (Luft, 1971). It has been extensively used as an inhibitor of diverse cation channels, including numerous TRP channels, select members of the $K_{2P}$ (KCNK) family, CALHM calcium channels, ryanodine receptors, and Piezo channels (Choi et al, 2019; Coste et al, 2012; Czirjak and Enyedi, 2003; Nagata et al, 2005; Pessah et al, 1985; Pope et al, 2020). The RR cation has a $6^+$ total charge and contains an essentially linear N-Ru-O-Ru-O-Ru-N backbone with coordinated ammonia groups completing the octahedra about three ruthenium atoms (Carrondo et al, 1980).

The structural mechanism of RR inhibition has been previously investigated for three channels: CALHM2 (Choi et al, 2019), mutant $K_{2P}$2.1(TREK-1)I110D (Pope et al, 2020) and TRPV6 (Neuberger et al, 2021). In the undecameric CALHM channel, RR functions as an antagonist rather than a pore blocker; the proposed mechanism of inhibition included dramatic conformational changes in response to RR binding near the pore-lining helix S1 (Choi et al, 2019). In the potassium channel $K_{2P}$2.1(TREK-1)I110D, a single RR molecule was observed bound directly on top of the SF through ionic interactions with the negatively charged Asp110 where it functions as a pure blocker (Pope et al, 2020). In a recent TRPV6 structure, a single RR molecule occluded the SF and closed the lower gate (Neuberger et al, 2021). Yet, it did not explain how RR binding at the SF could cause a gating transition in the lower pore. Additionally, it has been proposed that RR blocks

[1]Department of Systems Pharmacology and Translational Therapeutics, Perelman School of Medicine, University of Pennsylvania, Philadelphia, PA 19104, USA. [2]Institute for Neurophysiology, Hannover Medical School, 30625 Hannover, Germany. [3]Department of Physiology and Convergence Medical Science, Institute of Health Sciences, Gyeongsang National University Medical School, Jinju, Korea. [4]Department of Pharmacology, Physiology and Neuroscience, New Jersey Medical School, Rutgers University, Newark, NJ 07103, USA. [5]These authors contributed equally: Ruth A Pumroy, José J De Jesús-Pérez, Anna D Protopopova. ✉E-mail: vmb@pennmedicine.upenn.edu

non-selective $Ca^{2+}$ channels TRPV1-TRPV4 by interacting with an acidic residue at the top of the SF (Chung et al, 2005; Garcia-Martinez et al, 2000; Voets et al, 2002).

To further examine the mechanism of RR pore block in two different subgroups of TRPV channels and understand the mode of action of RR, we have chosen TRPV2 (non-selective $Ca^{2+}$ cation channel) and TRPV5 ($Ca^{2+}$-selective cation channel) for our studies. Both channels have already been well characterized structurally, revealing the expected flexible TRPV2 SF and rigid TRPV5 SF (Dang et al, 2019; Dosey et al, 2019; Fluck et al, 2022; Hughes et al, 2019, 2018a, 2018b; Pumroy et al, 2022, 2019; Zhang et al, 2022; Zubcevic et al, 2019, 2018b). Here, we present structures of each channel bound to RR either by itself, or in the presence of a known activator. Our data revealed that RR can enter the SF of both channels in their apo states, yet in the presence of activators RR can also alter protein conformation to further block the channels. In the presence of the TRPV2 activator 2-aminoethoxydiphenyl borate (2-APB) (Hu et al, 2004), RR is tightly coordinated in the SF and prevents the channel from entering the inactivated state. Similarly, in the presence of the TRPV5 activator phosphatidylinositol 4,5-bisphosphate $PI(4,5)P_2$ (Lee et al, 2005), RR blocks the entrance of $Ca^{2+}$ ions to the pore by binding in the SF below Asp542 and induces conformational changes in S6 to close the lower gate. Therefore, our data highlight the interplay between channel gating and RR block in both subfamilies of TRPV channels.

## Results and discussion

### RR-bound structures of TRPV2 with and without 2-APB activation

First, we performed whole-cell patch clamp recordings on HEK293T cells transiently expressing rat TRPV2. We observed that RR-induced inhibition of rTRPV2 was strongly voltage dependent, with potent inhibition at negative voltages and reduced inhibition at positive voltages (Figure EV1). Consequently, the activation of rTRPV2 was accomplished by holding cells at −60 mV and applying 0.5 mM 2-APB, after which the channel was exposed to varying concentrations of RR from the extracellular side (Figure EV1). Our results exhibited a concentration-dependent inhibition, yielding an $IC_{50}$ of $159 \pm 47$ nM (Figure EV1).

Next, we prepared cryo-EM grids of wild-type rat TRPV2 reconstituted into nanodiscs in the presence of 1 mM of the robust TRPV2 activator 2-APB and 1 mM RR. While RR is a strong inhibitor at negative membrane potential, it is unclear how tightly it binds in the absence of voltage. Therefore, we used a high concentration of RR to ensure saturation. This dataset yielded a single stable high-quality state: 2-APB-activated RR-blocked TRPV2 ($TRPV2_{2APB+RR}$) with C4 symmetry at 2.9 Å resolution (Appendix Table S1, Appendix Figs. S1 and S2). The strong extra density at the $TRPV2_{2APB+RR}$ SF was clearly visible from early stages of data processing before application of any symmetry and is aligned along the central symmetry axis (Appendix Fig. S2); the size and shape of this density is consistent with the expected dimensions for RR (Fig. 1A, Appendix Fig. S4) and density consistent with 2-APB is present at our previously proposed 2-APB binding site and not in the vanilloid pocket (Fig. 1B, Appendix Figs. S3 and S4; Pumroy et al, 2022). The SF of $TRPV2_{2APB+RR}$ has adapted to

accommodate RR, with tight coordination by the backbone carbonyl of Gly606 along the interior of the SF (Fig. 1A,C,D) and the lower gate is closed at Met645 (Fig. 1C,D).

We also prepared cryo-EM grids of TRPV2 in the presence of 1 mM RR without the activator. The dataset yielded a single stable state with unique and strong density at the SF: RR-bound TRPV2 ($TRPV2_{RR}$) in C1 at 3.3 Å resolution with a wide SF and closed lower gate (Figure EV2, Appendix Table S1, Appendix Figs. S1, S2, and S4). The $TRPV2_{RR}$ structure resembles the previously solved $TRPV2_{Apo2}$ state (Pumroy et al, 2019) (PDB 6U86, r.m.s.d. 0.643 Å), which was a minor state in the absence of activators with the same wide SF and closed lower gate (Figure EV2; Pumroy et al, 2019). The presence of RR stabilized the $TRPV2_{RR}$ SF in the $TRPV2_{Apo2}$-like state but did not reveal the expected shape for the RR molecule (Figure EV2, Appendix Fig. S4). The $TRPV2_{RR}$ SF harbors a large asymmetric density coordinated by Glu609 and the Gly606 backbone carbonyls from only some of the TRPV2 monomers (Figure EV2, Appendix Fig. S4). As this density is not present in the otherwise similar $TRPV2_{Apo2}$ (PDB 6U86/EMD-20678) structure, we attribute this density to RR, although the irregular shape does not allow for the molecule to be built. This irregular shape may be due to the flexibility of the TRPV2 SF, giving RR the freedom to sample multiple binding conformations. Thus, our results agree with the previous observation that the TRPV1-TRPV3 SF is dynamic and can permit cation passage even when the pore is closed at the lower gate (Jara-Oseguera et al, 2019).

### RR-bound structures of TRPV5 with and without PI(4,5)$P_2$ activation

The TRPV5 activator $PI(4,5)P_2$ is a native component of the plasma membrane, making TRPV5 constitutively active. To observe the effect of RR on TRPV5, we used whole-cell patch-clamp electrophysiology recordings on HEK293 cells transiently transfected with rabbit TRPV5. Consistent with earlier results (Nilius et al, 2001b), application of RR inhibited monovalent TRPV5 currents elicited by the removal of extracellular $Mg^{2+}$ and $Ca^{2+}$ with an $IC_{50}$ of $13.7 \pm 4.4$ nM (Figure EV3).

Next, we prepared cryo-EM grids of wild-type rabbit TRPV5 reconstituted into nanodiscs treated first with 400 μM $PI(4,5)P_2$ for 5–10 min and then 1 mM RR for 45 min before preparing grids. This dataset yielded a single stable state: $TRPV5_{PIP2+RR}$ with C4 symmetry at 2.65 Å resolution (Appendix Table S1, Appendix Figs. S5 and S6). $TRPV5_{PIP2+RR}$ contained strong extra density at the TRPV5 SF consistent with the size and shape of RR (Fig. 2A, Appendix Figs. S7 and S8), but did not show the expected density for $PI(4,5)P_2$ (Appendix Fig. S9). Notably, in our experience the addition of $PI(4,5)P_2$ to TRPV5 consistently yields datasets with a majority of particles with an open lower gate and extra density for $PI(4,5)P_2$ (Fluck et al, 2022; Hughes et al, 2018b; Lee et al, 2023) but none similar to the RR in the pore (Fig. 2C). To determine whether RR occupies the SF in the apo state, we collected a dataset of TRPV5 treated with 1 mM RR, yielding a single state with strong density at the SF with a C4 symmetry at 2.96 Å resolution (Fig. 2B, Appendix Table S1, Appendix Figs. S5 and S6).

Upon comparison between the TRPV5 pores (Fig. 2E–H), the addition of RR to the open channel led to the pore reverting to the

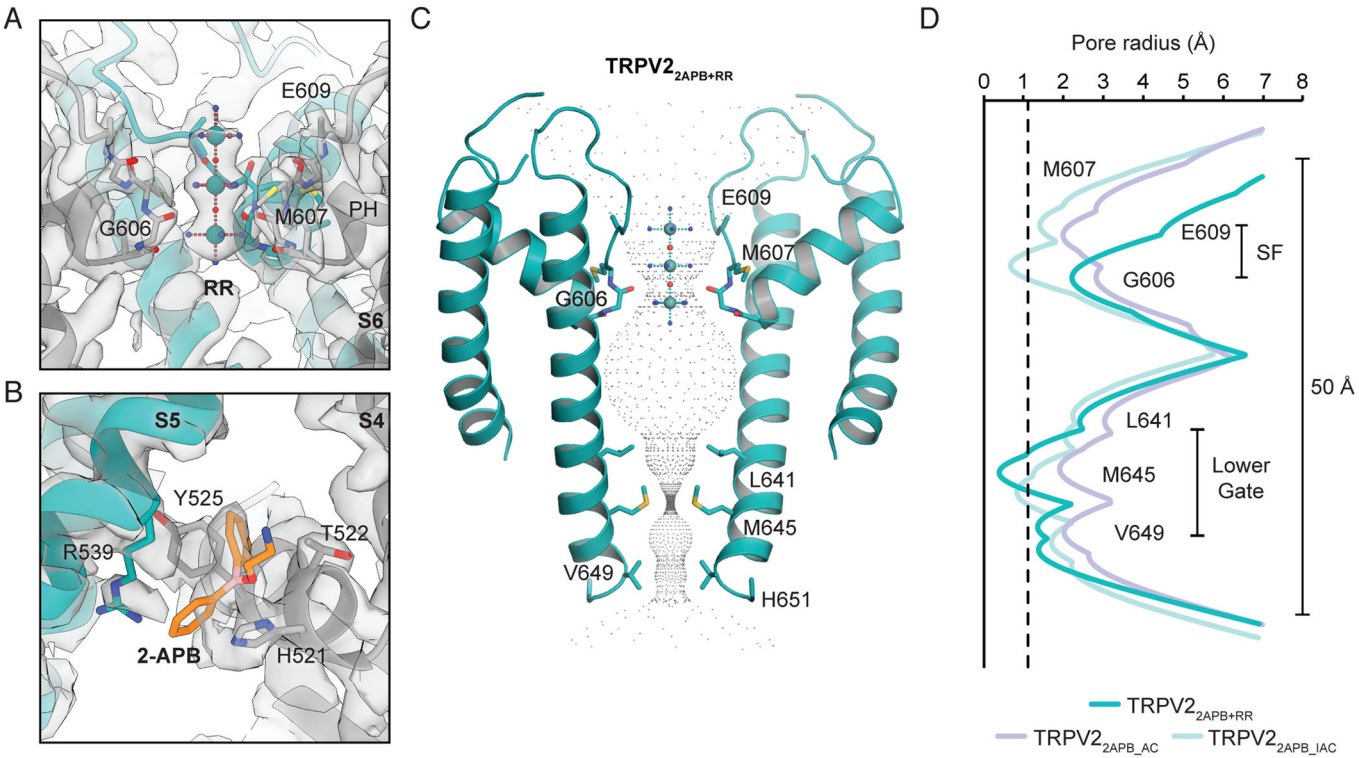

**Figure 1. RR pore block of the 2-APB-activated TRPV2.**

(A, B) Map density and cartoon representation of the RR (A) and 2-APB (B) binding sites in TRPV2$_{2\text{-APB+RR}}$. One monomer is colored teal, the adjacent monomers are colored gray; The map is countered at $\sigma = 5$ at the RR site and at $\sigma = 3.5$ at the 2-APB site. (C) HOLE-generated pore profile of TRPV2$_{2\text{-APB+RR}}$. (D) Graphical representation of the pore profiles of TRPV2$_{2\text{APB+RR}}$ (teal), TRPV2$_{2\text{APB\_AC}}$ (light blue) and TRPV2$_{2\text{APB\_IAC}}$ (pale cyan). The radius of a dehydrated calcium ion is marked by a dotted gray line at 1.1 Å.

closed conformation, identical to apo TRPV5 (Fluck et al, 2022) (PDB 7T6J, TRPV5$_{\text{Apo}}$, r.m.s.d. 0.445 Å) (Fig. 2D). TRPV5$_{\text{RR}}$ is also in an identical conformation to the previously solved TRPV5$_{\text{Apo}}$ state (r.m.s.d. 0.386 Å) (Fig. 2G). The only difference between these states is a large additional density for RR in the SF where bound ions were previously observed in TRPV5 (Fluck et al, 2022; Fig. 2A,B,D).

There appear to be two potential RR binding positions in the TRPV5 SF, either of which fully blocks the pore. TRPV5$_{\text{PIP2+RR}}$ also has density coordinated at Asp542 which is significantly separated from the RR density and is likely caused by the coordination of an ion at this position. Similar density can be found in TRPV5$_{\text{Apo}}$ and TRPV5$_{\text{PIP2}}$ but is absent in TRPV5$_{\text{RR}}$ as the site is occupied by RR (Fig. 2, Appendix Figs. S7 and S8).

## Comparison of RR binding sites in TRPV2 and TRPV5

The primary site for RR stabilization in TRPV2$_{2\text{APB+RR}}$ is the carbonyl of Gly606, which at distances of 3.0–3.2 Å can form hydrogen bonds with ammonia groups coordinated to the central and bottom ruthenium atoms (Fig. 3A). The next closest potential interacting group is the carbonyl of Gly608, which at 4.9–5.0 Å away from ammonia groups is likely too far away to coordinate RR (Fig. 3A). Additionally, density for the top ruthenium atom was more poorly resolved than the lower two ruthenium atoms, indicating that the top ruthenium group is not tightly coordinated

(Fig. 1A). Glu609 is close enough that the sidechain should be able to coordinate with RR, yet density for the sidechain was not resolved in this structure and could not be built (Fig. 1A).

In our efforts to probe this binding site, we chose not to try to alter the backbone carbonyl of Gly606 as these experiments are not trivial to perform and validate. Instead, we decided to mutate Glu609 as mutation of the analogous residue has been shown to alleviate RR inhibition in TRPV1, TRPV3, and TRPV4 (Chung et al, 2005; Garcia-Martinez et al, 2000; Voets et al, 2002). Additionally, we mutated Met607, which is not able to coordinate with RR but must move out of the path of the pore for it to open wide enough for RR to bind. We performed whole-cell patch clamp recordings on HEK293T cells transiently expressing the rat TRPV2 mutants Glu599Gln and Met607Ile, then compared the degree of 2-APB current inhibition after application of RR to the wild-type channel. Both Glu609Gln and Met607Ile showed a robust reduction in RR inhibition, with IC$_{50}$ values of $149 \pm 111$ and $29 \pm 5\ \mu M$, respectively (Figure EV4).

In comparison to TRPV2, TRPV5 is a Ca$^{2+}$ selective channel with a rigid SF that coordinates RR differently. The RR densities we observe in TRPV5$_{\text{PIP2+RR}}$ and TRPV5$_{\text{RR}}$ are similar in dimension to the RR density recently reported for TRPV6 (Neuberger et al, 2021). While the position of RR in TRPV5$_{\text{RR}}$ is identical to the position RR occupies in TRPV6, the RR in TRPV5$_{\text{PIP2+RR}}$ is positioned notably lower in the SF (Fig. 2A,B, Appendix Fig. S10). This reflects the three Ca$^{2+}$ ion binding positions proposed at the

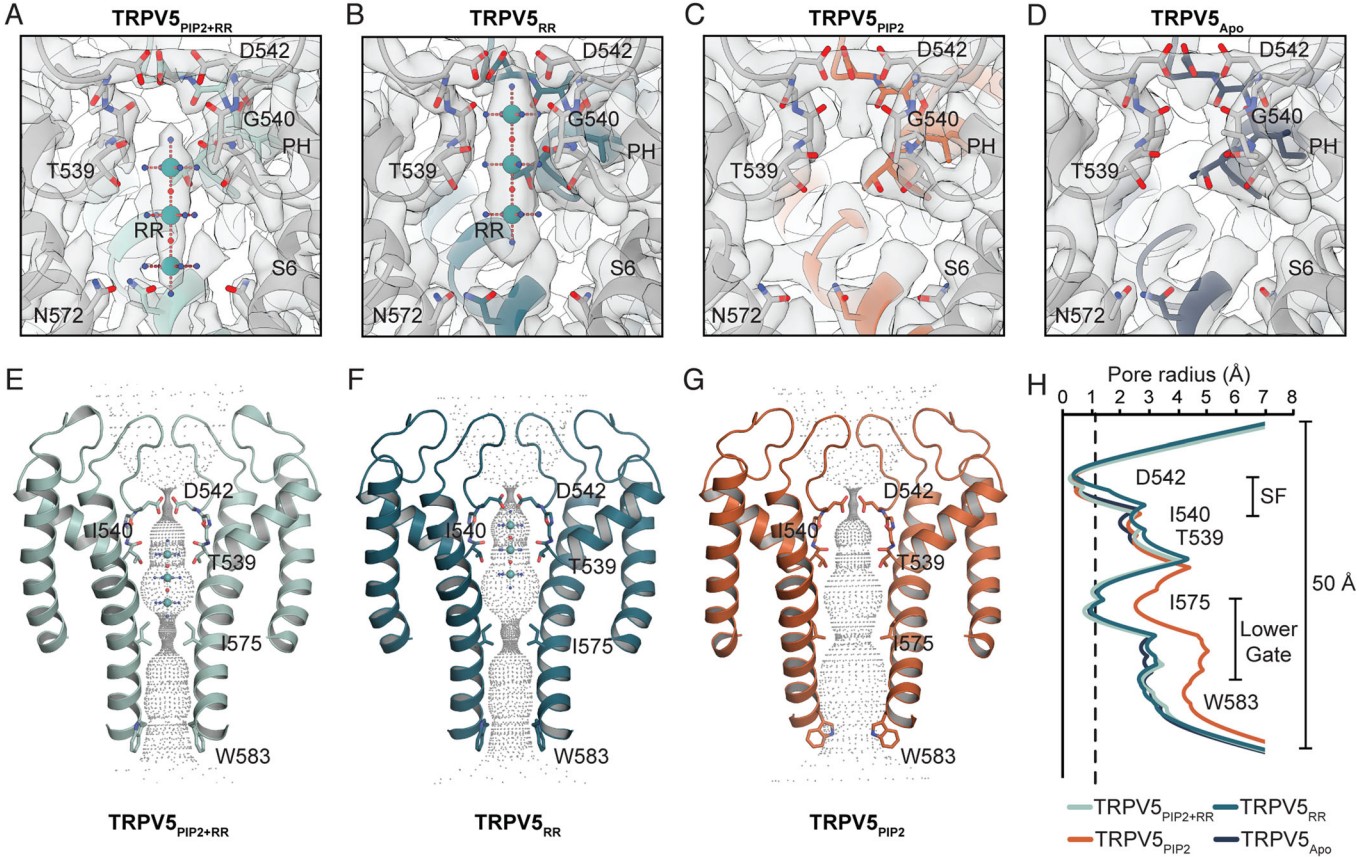

**Figure 2. TRPV5 pore block by RR in the presence and absence of the activator PI(4,5)P₂.**

(A–D) Map density and cartoon representation of the RR binding site in TRPV5$_{PIP2+RR}$ (A), TRPV5$_{RR}$ (B), TRPV5$_{PIP2}$ (EMD-29049, PDB 8FFO) (C), and TRPV5$_{Apo}$ (EMD-25716, PDB 7T6J) (D). One monomer is colored blue or orange, the adjacent monomers are colored gray. Maps contoured at $\sigma = 5$. (E–G) HOLE-generated pore profiles of TRPV5$_{PIP2+RR}$ (E), TRPV5$_{RR}$ (F), and TRPV5$_{PIP2}$ (G). (H) Graphical representation of the pore profiles of TRPV5$_{PIP2+RR}$ (light blue), TRPV5$_{RR}$ (medium blue), TRPV5$_{PIP2}$ (orange), and TRPV5$_{Apo}$ (dark blue). The radius of a dehydrated calcium ion is marked by a dotted gray line at 1.1 Å.

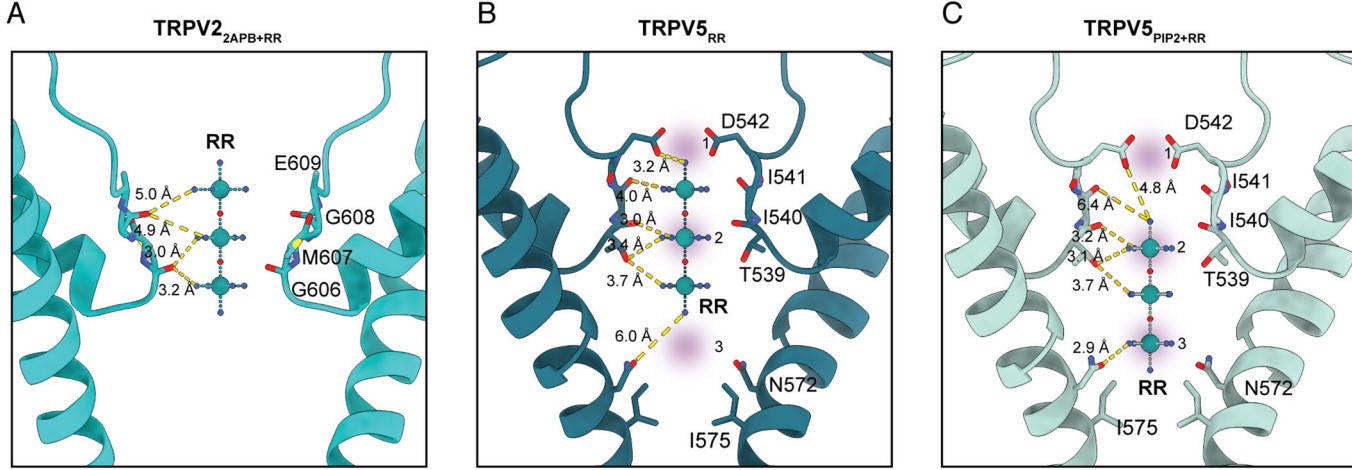

**Figure 3. RR coordination in the selectivity filters of TRPV2 and TRPV5.**

(A–C) RR coordination in TRPV2$_{2-APB+RR}$ (A), TRPV5$_{RR}$ (B), and TRPV5$_{PIP2+RR}$ (C). Distances to nearby polar groups are marked with yellow dashed lines. In (B, C), the proposed Ca$^{2+}$ binding sites are indicated by purple shading.

SF in TRPV6 (Saotome et al, 2016; Singh et al, 2017), with the observed RR binding in two different positions spanning across all three proposed $Ca^{2+}$ binding sites in the SF. The RR molecule in TRPV5$_{RR}$ is coordinated high in the SF, occupying Sites 1 and Site 2 (Fig. 3B). Asp542 forms a polar interaction (3.2 Å) with the top ammonia group (Site 1) and the carbonyl of Thr539 and possibly also the Thr539 sidechain hydroxyl form hydrogen bonds (3.0–3.4 Å) with ammonia groups coordinated with the middle ruthenium atom (Site 2) (Fig. 3B). The RR molecule in TRPV5$_{PIP2+RR}$ remains coordinated at Site 2, with the Thr539 backbone carbonyl and sidechain hydroxyl forming hydrogen bonds (3.1–3.2 Å) with ammonia groups of the top ruthenium atom (Fig. 3C). The bottom of the RR molecule now occupies Site 3, which allows for the formation of hydrogen bonds with Asn572 (2.9 Å) (Fig. 3C).

To validate the contact residues within the binding site of RR, we generated a single mutation (T539A) and a double mutation (T539A/D542A) to investigate whether these mutants exhibit reduced inhibition by RR compared to wild type TRPV5 channels in whole-cell patch clamp experiments (Figure EV5). The application of 1 µM RR resulted in a 77.3 ± 2.9% inhibition in wild type TRPV5 channels. The T539A mutant showed a reduced, but highly variable 44.7 ± 14.7% inhibition that was not significantly different from wild type ($p = 0.1$). The double mutant T539A-D542A was not inhibited by 1 µM RR. Application of 10 µM RR inhibited the wild type and T539A mutants to a similar extent, 96.7 ± 1.2% and 94.5 ± 1.5%, respectively. The double mutant T539A-D542A was not inhibited by 10 µM RR. These data show that the D542 residue is a key determinant of RR inhibition of TRPV5. Notably, the D542A mutation was also shown previously to eliminate $Ca^{2+}$ permeation, $Mg^{2+}$ block and inward rectification of TRPV5 (Nilius et al, 2001a).

## Comparison of RR-induced conformational changes in TRPV2 and TRPV5

In our previously published work on 2-APB activation of TRPV2, we observed two states: a minor activated state (TRPV2$_{2APB\_AC}$, PDB 7N0N) and a dominant inactivated state (TRPV2$_{2APB\_IAC}$, PDB 7N0M) (Pumroy et al, 2022). We proposed that 2-APB binding to TRPV2 drives activation of the channel by functioning as a wedge to induce a 7° counterclockwise rotation of the VSLDs and ARDs (viewed from the extracellular side), which is a consistent movement seen across TRPV1-TRPV3 channel opening (Kwon et al, 2022; Neuberger et al, 2022; Zhang et al, 2021). Conversely, we observed that a 7° clockwise rotation of the same domains is necessary to take on the inactivated state. Intriguingly, preparing TRPV2 with a combination of 2-APB and RR stabilized the TRPV2$_{2APB+RR}$ structure in an activated-like conformation (r.m.s.d. 0.743 compared to TRPV2$_{2APB\_AC}$, r.m.s.d. 2.166 compared to TRPV2$_{2APB\_IAC}$) (Fig. 4A,B), while the inactivated state is the most stable conformation when TRPV2 is treated with only 2-APB (Pumroy et al, 2022). The presence of RR in the SF locks it in a more open conformation, which keeps the pore domain from fully rotating inwards (Fig. 4A). The only significant conformational difference between TRPV2$_{2APB+RR}$ and TRPV2$_{2APB\_AC}$ is at the pore, where a downward rotation (9°) and shift (1.1 Å) of the pore helix to accommodate RR leads to subtle shifts in S5 and S6

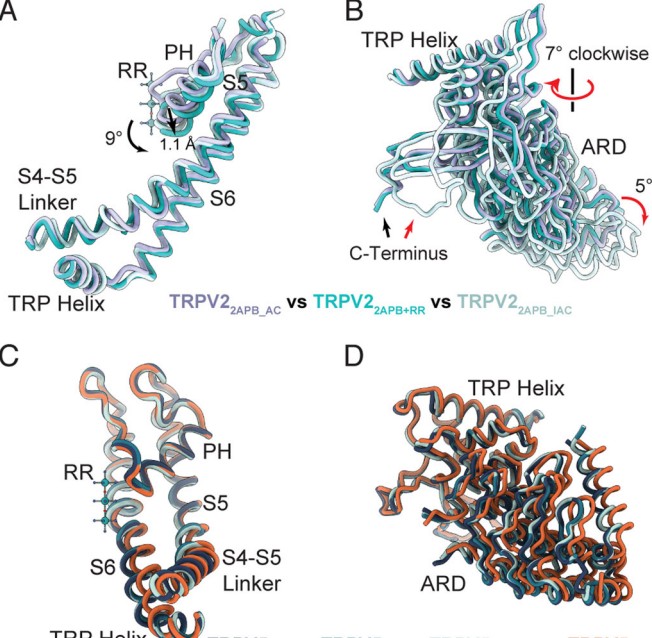

**Figure 4. Conformational changes of TRPV2 and TRPV5 induced by RR block.**

(A, B) Cartoon representations comparing the pore domain (A) and ARD (B) movements between TRPV2$_{2APB\_AC}$ (purple), TRPV2$_{2APB+RR}$ (teal), and TRPV2$_{2APB\_IAC}$ (light cyan). Models were aligned to the pore domain. Movements between TRPV2$_{2APB\_AC}$ and TRPV2$_{2APB+RR}$ are indicated by black arrows. Movements between TRPV2$_{2APB+RR}$ and TRPV2$_{2APB\_IAC}$ are indicated by red arrows. (C, D) Cartoon representations comparing the pore domain (C) and ARD (D) movements between TRPV5$_{PIP+RR}$ (light blue), TRPV5$_{RR}$ (medium blue), TRPV5$_{PIP2}$ (EMD-29049, PDB 8FFO) (orange), TRPV5$_{Apo}$ (EMD-25716, PDB 7T6J) and (dark blue).

which constrict the lower gate at Met645 (Figs. 1C,D and 4A). The rest of the channel maintained the previously observed activated conformation, where the C-terminus is in a helical conformation rather than the loop conformation seen in the inactivated state (Fig. 4B; Pumroy et al, 2022). This suggests that the switch to the inactivated C-terminal loop may depend on a confirmation that is destabilized when the SF is not tightly closed. The addition of RR stabilized the activated-like conformation of TRPV2, which in the absence of RR was an ephemeral state (Pumroy et al, 2022). As a result, density for lipids and 2-APB are more clearly resolved and are consistent with what we predicted previously: clear three-lobed density for 2-APB coordinated by His521 and Arg539 (Fig. 1B) and minimal density, whether for lipids or 2-APB, in the vanilloid pocket in comparison to clear vanilloid lipid density in TRPV2$_{RR}$ (Appendix Fig. S3).

Conversely, the addition of RR to PI(4,5)$P_2$-activated TRPV5 caused the channel to close and take on a conformation identical to TRPV5$_{Apo}$ (Fig. 4C,D). No density in the TRPV5$_{PIP2+RR}$ map was observed for the PI(4,5)$P_2$ molecule (Appendix Fig. S9), likely because the channel is closed. Indeed, the structures of both TRPV5$_{RR}$ and TRPV5$_{PIP2+RR}$ are essentially identical to TRPV5$_{Apo}$, including vanilloid lipid densities (Appendix Fig. S9), apart from the density for RR (Fig. 4C,D).

## Interrelation between RR pore block and channel gating in TRPV channels

RR is the most universal and robust pore blocker used to study TRPV channels (Seebohm and Schreiber, 2021). Recently, a RR binding site in the SF of the $Ca^{2+}$-selective TRPV6 channel has been revealed, but that structure revealed conformational changes that could not be explained (Neuberger et al, 2021). This introduced the possibility of an interplay between RR pore block and channel gating in TRPV channels, which remains largely unexplored. Here, we present a molecular view of RR block of non-selective $Ca^{2+}$ channel TRPV2 and $Ca^{2+}$-selective channel TRPV5 upon activation of both channels.

Our TRPV2$_{2APB+RR}$ structure revealed that binding of RR to the 2-ABP-activated TRPV2 stabilized the channel in an intermediate conformation, preventing ion flow through the occluded SF and closing the lower gate at Met645, but at the same time inhibiting the transition to the inactivated state. This TRPV2 locked in an activated conformation allowed us to resolve clear density for 2-APB in our previously proposed 2-APB binding site (Pumroy et al, 2022), but not in the vanilloid pocket, a recently suggested alternate 2-APB site in mouse TRPV2 (Su et al, 2022). The observed allosteric link between RR binding at the SF and conformational changes at the lower gate fit well with previous evidence for allostery in TRPV1–TRPV4 (Hilton et al, 2019; Kwon et al, 2022; Zhang et al, 2021).

For TRPV5 and TRPV6, the allosteric link between the SF and the lower gate has not been clearly defined. Application of RR to PI(4,5)P$_2$-activated TRPV5 caused the channel to close. Based on our data, we propose that RR interaction with Asn572 at Site 3 (Fig. 3C) may pull the S6 helix towards the central axis of the pore and closes the lower gate at Ile575. This closure would draw the key Arg584 residue away from the PI(4,5)P$_2$ binding site (Hughes et al, 2018b), leading to the observed loss of PI(4,5)P$_2$ binding (Appendix Fig. S9). We also observed two possible positions for RR binding in the TRPV5 SF, a higher one which seems to be preferred when the channel has not been opened coordinated at Asp542 and a lower one coordinated by Asn572 which is dominant when the channel has previously been activated by PI(4,5)P$_2$. We propose that RR, in its attempt to traverse the pore like any other cation, encounters obstacles due to interaction with the Asn572 and its physicochemical properties, including charge, and size. These characteristics collectively impede its movement within the permeation pathway, causing blockage of the channel and consequent closure.

We observed that the RR molecule can bind inside the pores of both channels even without activation. TRPV2 has a flexible and adaptable SF which can easily allow for the entrance of the bulky RR while the channel is in the TRPV2$_{Apo2}$ conformation with the open SF. Yet RR is not tightly coordinated in this position, resulting in somewhat amorphous density. It is not clear how RR enters the TRPV5 pore which is tightly closed at both gates in the TRPV5$_{Apo}$ state. There are two possible entrances to the TRPV5 channel vestibule: through the SF or the lower gate. In TRPV5$_{Apo}$, the lower gate is tightly closed and hydrophobic and the SF is also closed, but provides a favorable environment for the entrance and binding of the positively charged molecule. Therefore, it is most likely that RR enters the TRPV5 pore through the SF, suggesting that the TRPV5 SF may not be not as rigid as commonly accepted. In the recently published structure of RR-bound TRPV6 (PDB 7S8B) RR was also observed inside the pore without an added activator (Neuberger et al, 2021), but in that case the apo TRPV6 structure (PDB 7S89) was resolved with an open lower gate providing a potential entrance to the pore vestibule for the RR molecule.

A clear commonality in RR binding to both TRPV2 and TRPV5 was revealed by the comparison of the structures presented here: backbone carbonyls of the SF appear to be a critical RR coordination site for channel block. In both TRPV2$_{2APB+RR}$ and TRPV5$_{PIP2+RR}$, the density for RR was strongest at Gly607/Thr539 and TRPV5$_{RR}$ showed strong density at this position as well (Figs. 1A and 2A,B). This points to how RR can block such a wide range of cation channels as the SFs of many cation channels depend on backbone carbonyl coordination.

Our studies show on a molecular level how RR can effectively block the 2-APB-activated non-selective $Ca^{2+}$ channel TRPV2 and the PI(4,5)P$_2$-activated $Ca^{2+}$ selective channel TRPV5. This new information reveals the interplay between block and activation in TRPV channels and will help advance the development of TRPV channel inhibitors.

# Methods

## Reagents and tools

See Table 1.

**Table 1. Reagents and tools.**

| Reagent/resource | Reference or source | Identifier or catalog number |
|---|---|---|
| Experimental models | | |
| Human Embryonic Kidney 293 (HEK293) cells | ATCC | Cat# CRL-1573 |
| Human Embryonic Kidney 293 (HEK293T) cells | ATCC | Cat# CRL-3216 |
| *Saccharomyces cerevisiae* BJ5457 | ATCC | Cat# 208282 |
| Recombinant DNA | | |
| YepM rat TRPV2 plasmid | Huynh et al 2014 | N/A |
| rat TRPV2 plasmid | Caterina et al 1999 | N/A |

**Table 1.** (continued)

| Reagent/resource | Reference or source | Identifier or catalog number |
|---|---|---|
| YepM rabbit TRPV5 plasmid | Hughes et al, 2018a | N/A |
| Rabbit TRPV5-IRES-GFP plasmid | Hughes et al, 2019 | N/A |
| pMSP2N2 | Grinkova et al, 2010, Addgene | Cat# 29520 |
| Antibodies | | |
| 1D4 primary antibody | Hodges et al 1988 | N/A |
| Oligonucleotides and sequence-based reagents | | |
| TRPV5 T539A Forward primer: CTGTTCCTCGCCATCATCGACGGC | Bionics | N/A |
| TRPV5 T539A Reversed primer: GATGATGGCGAGGAACAGCTCAAAGGTGC | Bionics | N/A |
| TRPV5 T539A/D542A Forward primer: TTCCTCGCCATCATCGCCGGCCCTGC | Bionics | N/A |
| TRPV5 T539A/D542A Reversed primer: GGGCCGGCGATGATGGCGAGGAACAGC | Bionics | N/A |
| rTRV2 M607I Forward primer: GCCAGCTCCCCTATACCAATGGTGAACTTGAAC | Eurogentec | N/A |
| rTRPV2 M607I Reversed primer: GTTCAAGTTCACCATTGGTATAGGGGAGCTGGC | Eurogentec | N/A |
| rTRPV2 E609Q Forward primer: CTGGAAAGCCAGCTGCCCCATACCAATGG | Eurogentec | N/A |
| rTRPV2 E609Q Reversed primer: CCATTGGTATGGGG CAGCTGGCTTTCCAG | Eurogentec | N/A |
| Chemicals, enzymes, and other reagents | | |
| SD-Leu Media | Fisher BoiReagents | Cat# MP114811075 |
| Glycerol | Fisher BioReagents | Cat# BP229-4 |
| Protease inhibitor cocktail | Sigma-Aldrich | Cat# P8215 |
| CnBr-activated sepharose beads | Cytiva | Cat# 17043001 |
| 1D4 peptide | Genscript | N/A |
| Lauryl Maltose Neopentyl Glycol (LMNG) | Anatrace | Cat# NG310 |
| Decyl Maltose Neopentyl Glycol (DMNG) | Anatrace | Cat# NG322 |
| TCEP | Pierce | Cat# PG82090 |
| Soy polar lipids | Avanti | Cat# 541602C |
| Bio-Beads SM-2 Absorbent media | BioRad | Cat# 1528920 |
| diC8-PI(4,5)P2 | Echelon Biosciences | Cat# P-4508 |
| Ruthenium red | Sigma-Aldrich | Cat# 2751 |
| Ruthenium red | Calbiochem | Cat# 557450 |
| Ruthenium red | TOCRIS | Cat# 1439 |
| jetPEI | Polyplus-transfection | Cat# 13-101-10N |
| Effectene Transfection Reagent | Qiagen | Cat# 301425 |
| Poly-L-lysine | Sigma-Aldrich | Cat# P4707 |
| Dulbecco's modified Eagle medium | Gibco | Cat# 11995-065 |

**Table 1.** (continued)

| Reagent/resource | Reference or source | Identifier or catalog number |
|---|---|---|
| Fetal bovine serum | Gibco | Cat# 12483-020 |
| Fetal bovine serum | Biochrom | Cat# F2442 |
| Software | | |
| pClamp 11.1 | Molecular Devices | https://www.moleculardevices.com/products/axon-patch-clamp-system/acquisition-and-analysis-software/pclamp-software-suite |
| Patchmaster | HEKA Electronik | https://www.heka.com/ |
| Relion 4.0 | Kimanius et al, 2021 | https://github.com/3dem/relion |
| cryoSPARC v3.3.x and v4.0.x | Punjani et al, 2021; Punjani et al, 2020; Punjani et al, 2017 | https://cryosparc.com |
| COOT | Emsley et al, 2004 | https://www2.mrc-lmb.cam.ac.uk/personal/pemsley/coot |
| ISOLDE | Croll, 2018 | https://tristanic.github.io/isolde/ |
| eLBOW | Moriarty et al, 2009 | https://phenix-online.org/documentation/reference/elbow_gui.html |
| PHENIX | Afonine et al, 2018 | https://phenix-online.org/documentation/index.html |
| HOLE | Smart et al, 1996 | http://www.holeprogram.org/ |
| Chimera | Pettersen et al, 2004 | https://www.cgl.ucsf.edu/chimera/ |
| ChimeraX | Goddard et al, 2018; Pettersen et al, 2021 | https://www.cgl.ucsf.edu/chimerax/ |
| PyMol 2.3 | Schrodinger, Inc. | https://pymol.org/2/ |
| Origin2021 | Origin Lab | https://www.originlab.com/2021 |
| Origin 8.5.1 | Origin Lab | https://www.originlab.com/ |
| Excel | Microsoft | https://www.microsoft.com/en-us/microsoft-365/excel |
| Other | | |
| Alkali-Cation Yeast Transformation Kit | MP Biomedicals | Cat# 112200200 |
| QuikChange II XL Site Directed Mutagenesis Kit | Agilent | Cat# 200522 |
| Quantifoil 1.2/1.3 grid | Quantifoil Micro Tools | Cat# Q3100CR1.3 |
| Superdex 200 column | Cytiva | Cat# 28990944 |

## Expression and purification

Wild-type rat TRPV2 and rabbit TRPV5 were prepared as previously described (Fluck et al, 2021). TRPV2 and TRPV5 cloned into a YepM vector and tagged with a C-terminal 1D4 epitope were expressed in BJ5457 *Saccharomyces cerevisiae* (ATCC) grown in SD-Leu media (MP Biomedicals) at 30 °C with 200 rpm shaking. Cells were harvested at an $OD_{600}$ 1.0–1.4 for maximum recombinant protein expression and resuspended in homogenization buffer (25 mM Tris-HCl, pH 8.0, 300 mM Sucrose, 5 mM EDTA, and a yeast protease inhibitor cocktail (Sigma)). Resuspended cells were lysed using a M110Y microfluidizer (Microfluidics). Cellular debris was removed via centrifugation at $14,000 \times g$ and then the membranes were pelleted by centrifugation

at $100,000 \times g$. Pelleted membranes were harvested and resuspended on ice using a tissue homogenizer (Kontes Duall with PTFE pestle) in buffer containing 25 mM Tris-HCl, pH 8.0, 300 mM Sucrose, and 1 mM PMSF. TRPV2 or TRPV5 were extracted from membranes in solubilization buffer (20 mM Hepes, pH 8.0, 150 mM NaCl, 5% glycerol for TRPV2 or 10% for TRPV5, 870 µM LMNG, 2 mM TCEP, and 1 mM PMSF). This mixture was clarified by centrifugation at $100,000 \times g$ and the insoluble fraction discarded. The solubilized TRPV2 or TRPV5 was bound to 1D4 antibody coupled CnBr-activated Sepharose beads (Cytiva), followed by washes with wash buffer (20 mM Hepes, pH 8.0, 150 mM NaCl, 2 mM TCEP) supplemented with 63 µM DMNG. The protein was eluted with wash buffer supplemented with 63 µM DMNG and 3 mg/mL 1D4 peptide. The purified TRPV2 or TRPV5

was concentrated by centrifugal ultrafiltration in an Amicon Ultra-15 Centrifugal Filter Unit with a 100 kDa cutoff (Sigma) to a volume under 1 mL and reconstituted into nanodiscs in a 1:1:200 molar ratio of TRPV tetramer:MSP2N2:soy polar lipids (Avanti). The soy polar lipids were rapidly dried under a nitrogen flow and further dried under vacuum before being resuspended in wash buffer containing DMNG in a 1:2.5 ratio (soy polar lipids:DMNG). The assembled nanodisc reconstitution mixture was incubated at 4 °C for 30 min before adding Bio-Beads to the mixture. After 1 h, the reconstitution mixture was transferred to a new tube with fresh Bio-Beads and incubated overnight at 4 °C. The nanodisc embedded TRPV2 or TRPV5 was purified from empty nanodiscs using a Superose 6 Increase 10/300 GL column (Cytiva) equilibrated in wash buffer for size-exclusion chromatography. The eluted TRPV2 or TRPV5 was concentrated by centrifugal ultrafiltration in an Amicon Ultra-2 Centrifugal Filter Unit with a 100 kDa cutoff (Sigma) to 2–2.57 mg/mL to use in vitrification. All purification steps were performed at 4 °C.

MSP2N2 was expressed and purified as previously described (Hughes et al, 2018b). MSP2N2 cloned into a pET28a vector (Addgene) was expressed in BL21 (DE3) cells. After harvest, the cells were resuspended in a buffer containing 20 mM Tris-HCl, pH 7.4, 1 mM PMSF, and a complete EDTA-free protease inhibitor cocktail tablet (Roche) and lysed using a M110Y Microfluidizer (Microfluidics). Cellular debris was removed via centrifugation at $14,000 \times g$ and the lysate supernatant was bound to Ni-NTA resin. The resin was washed with wash buffer (20 mM Tris-HCl, pH 7.4, 100 mM NaCl), then by wash buffer supplemented with 1% Triton X-100, then by wash buffer supplemented with 50 mM sodium cholate, and finally by wash buffer supplemented with 20 mM imidazole. MSP2N2 was eluted from the Ni-NTA resin with wash buffer supplemented with 300 mM imidazole and further purified by size-exclusion chromatography on a Superdex 200 column (Cytiva) equilibrated with 50 mM Tris-HCl, pH 7.5, 100 mM NaCl, and 5 mM EDTA before being concentrated to ~10 mg/mL.

## Cryo-EM sample preparation and data collection

For RR-bound TRPV2 or TRPV5, 1 mM RR was incubated with protein for 5 min on ice prior to grid preparation. For RR-bound activated TRPV2, protein was incubated 5 min on ice with 1 mM 2-APB and 1 mM RR. For RR-bound activated TRPV5, protein was incubated on ice for 45 min with 1 mM RR after 5–10 min incubation on ice with 400 µM diC8-PI(4,5)P$_2$ (Echelon Biosciences). 200 mesh Quantifoil 1.2/1.3 grid (Quantifoil Micro Tools) were glow discharged for 30 s at 10mAmp in a PELCO easiGlow system. For the TRPV2 sample only, 3 mM fluorinated Fos-choline 8 (Anatrace) was added immediately before blotting to improve particle distribution in vitreous ice. 3 µL of the final sample was applied to a freshly glow discharged grid and then blotted for 5–8 s with 0 blot force at 4 °C and 100% humidity before vitrification in liquid ethane in Vitrobot Mark IV (FEI).

Cryo-EM samples were imaged on a Titan Krios G3i 300 kV electron microscope with a Gatan K3 direct electron detector. Forty frame movies were collected with a nominal dose of 42 e$^-$/Å$^2$. Super-resolution images were collected at a magnification of either ×81,000 or ×105,000 for a pixel size of 0.535, 0.42, or 0.415 Å/pixel.

## Cryo-EM data processing

For the TRPV2$_{2APB+RR}$ dataset, 10,643 movies were collected and processed using both Relion 4.0 (Kimanius et al, 2021) and cryoSPARC v3.3 (Punjani et al, 2017). Movies were motion corrected using the Relion implementation of MotionCor2 (Zheng et al, 2017) and binned to a pixel size of 0.834. CTF values were determined for the micrographs using CTFFind4 (Rohou and Grigorieff, 2015) and the micrographs were then curated to remove suboptimal micrographs, resulting in 6887 good micrographs. Particles were picked using a Topaz model trained on the micrographs after Topaz denoising and then extracted with a particle threshold of −3 (Bepler et al, 2019). In all, 690,077 particles were extracted binned to a pixel size of 3.336 Å/pixel and subjected to 2D classification to remove false positives and bad particles. The best particles from the first round were re-extracted to a pixel size of 1.668 Å/pixel and subjected to another round of 2D classification, resulting in 516,889 particles in the best 2D classes. The particles were then reconstructed in a 3D classification job without applied symmetry using TRPV2$_{Apo1}$ (EMD-20677) as a reference. Three classes with a combined 201,369 particles showed density for transmembrane helices and yielded structures at 3.94 Å without symmetry or 3.58 Å with applied C4 symmetry. The particles refined with C4 symmetry were sorted by 3D classification without angular assignment, using a mask that covered the entire protein but excluding nanodisc density. One class composed of 65,741 particles emerged from that sorting with the highest resolution and well resolved density for the pore loops. These particles refined with C4 symmetry to a structure at 3.5 Å resolution. The particles were then re-extracted to their final pixel size of 0.834 Å/pixel and subjected to multiple rounds of 3D refinement with C4 symmetry, CTF refinement (Zivanov et al, 2018), and Bayesian polishing, yielding a structure at 3.1 Å before sharpening. The particles were then imported to cryoSPARC and subjected to non-uniform refinement (Punjani et al, 2020) and further CTF refinements, yielding a structure at 2.9 Å. The final map was produced by Phenix Resolve density modification (Terwilliger et al, 2020).

For the TRPV2$_{RR}$ dataset, 12,659 movies were collected and processed using Relion 4.0 (Kimanius et al, 2021). Movies were motion corrected using the Relion implementation of MotionCor2 (Zheng et al, 2017) and binned to a pixel size of 1.07 Å/pixel. CTF values were determined for the micrographs using CTFFind4 (Rohou and Grigorieff, 2015). Particles were picked using a Topaz model trained on the dataset and Topaz extracted with a particle threshold of −2 (Bepler et al, 2019). 590,270 particles were extracted binned to a pixel size of 4.28 Å/pixel and subjected to 2D classification to remove false positives and bad particles. The best particles from the first round were re-extracted to a pixel size of 2.14 Å/pixel and subjected to another round of 2D classification, resulting in 433,561 particles in the best 2D classes. The particles were then reconstructed in a 3D classification job without applied symmetry using TRPV2$_{Apo1}$ (EMD-20677) as a reference. One class with 139,961 particles showed good density for transmembrane helices and yielded a structure at 4.35 Å without symmetry. The particles were sorted by 3D classification without angular assignment, using a mask that covered the entire protein but excluding nanodisc density. Two classes with combined 55,445 particles had the best overall quality and after extracting to their final pixel size, 1.07 Å, yielded a structure at 4.03 Å. The particles were then

subjected to multiple rounds of CTF refinement (Zivanov et al, 2018) and Bayesian polishing, yielding a structure at 3.51 Å before sharpening and 3.47 Å with Relion post-processing.

For the TRPV5$_{RR}$ dataset, 9,237 movies were collected and processed using cryoSPARC v3.3 (Punjani et al, 2017). Movies were patch motion corrected with an alignment resolution of 3 Å and Fourier cropped to half the resolution. The micrographs were then run through patch CTF estimation. 2D templates for template picking were generated on a subset of 200 micrographs using the blob picker and 2D classification jobs. In all, 1,640,882 particles were picked, extracted with a box size of 288 and binned by a factor of 4. 2D classification was performed to remove false positives and bad particles, reducing the particle count to 131,133 particles. Re-extraction was performed on the remaining particles without binning in a box size of 288. An ab initio reconstruction with one class in C1 symmetry was generated from these particles and then subjected to a heterogeneous refinement with 3 classes in C1 symmetry. One class with 65,712 particles emerged that refined to higher resolution. These particles were then subjected to non-uniform refinement which gave a 3.3 Å structure (Punjani et al, 2020). An additional round of heterogeneous refinement with 3 classes gave one class with 39,976 particles that refined to higher resolution than the other classes. A final non-uniform refinement of these good particles yielded a structure at 2.96 Å with C4 symmetry (Punjani et al, 2020).

For the TRPV5$_{PIP22+RR}$ dataset, 12,412 movies were collected and processed using Relion 4.0 (Kimanius et al, 2021). Movies were motion corrected using the Relion implementation of MotionCor2 (Zheng et al, 2017). CTF values were determined for the micrographs using CTFFind4 (Rohou and Grigorieff, 2015). In all, 46,651 particles were autopicked using Laplacian-of-Gaussian blob detection from the subsets of 150 micrographs and used for Topaz training (Bepler et al, 2019). Particles were picked using the Topaz model trained on the entire dataset and then extracted with a particle threshold of −2 (Bepler et al, 2019). 6,859,887 particles were extracted binned to a pixel size of 3.32 Å/pixel and subjected to 2D classification to remove false positives and bad particles. 1,514,980 particles were extracted and subjected to a 3D classification job without applied symmetry using TRPV5$_{Apo}$ (EMD-25716) as a reference. One class with 439,366 particles was selected and extracted unbinned to a pixel size of 0.83 Å/pixel, then 3D refinement in C4 symmetry. These particles were subjected to CTF refinement (Zivanov et al, 2018) and Bayesian polishing, yielding a structure at 3.0 Å. A second 3D classification focusing on the pore (S5, S6) region were used to separate closed and open pore states. One well-defined class composed of 368,030 was emerged at a resolution of 2.7 Å. Another round of CTF refinements and Bayesian polishing yielded a structure at 2.6 Å in C4 symmetry before sharpening. The final map was produced by Phenix Resolve density modification (Terwilliger et al, 2020).

## Model building

For TRPV2$_{RR}$ and TRPV2$_{2APB+RR}$ the models of TRPV2$_{Apo2}$ (PDB 6U86) and TRPV2$_{2APB\_AC}$ (PDB 7N0M) were used as starting models and docked into the maps in Coot (Emsley and Cowtan, 2004). For TRPV5$_{RR}$, and TRPV5$_{RR+PIP2}$, the model of TRPV5$_{Apo}$ (PDB 7T6J) was used as a starting model and docked into the maps

in Coot (Emsley and Cowtan, 2004). The models were then iteratively manually adjusted to the map and refined using either phenix.real_space_refine from the PHENIX software package (Afonine et al, 2018) or ISOLDE (Croll, 2018). The ligand restraint files for RR were generated based on RR from PDB 6UIW using the eLBOW tool from the PHENIX software package (Moriarty et al, 2009). Pore profiles were generated using Hole (Smart et al, 1996). Images of the models and maps for figures were generated using Pymol (Schrödinger), Chimera (Pettersen et al, 2004), and ChimeraX (Goddard et al, 2018). Local resolution maps were calculated using the cryoSPARC implementation of BlocRes (Heymann and Belnap, 2007).

## TRPV2 electrophysiology

cDNA of rTRPV2 was a kind gift from Dr. Michael Caterina (Johns Hopkins University School of Medicine, Baltimore). The mutants rTRPV2-M607I and rTRPV2-E609I were generated by site directed mutagenesis with the Quikchange lightning site-directed mutagenesis kit (Aglient, Waldbronn, Germany) according to the instructions of the manufacturer. All mutants were sequenced subsequently to exclude further channel mutation and to prove intended amino acid exchange. HEK293T cells (ATCC) were transfected with rat TRPV2-plasmids using jetPEI (Polyplus-transfection ® SA, Illkirch, France). Nanofectin (PAA) was used to transfect HEK293T cells (ATCC) with rat TRPV2. Cells were cultured at 37 °C with 5% CO$_2$ in Dulbecco's modified Eagle medium nutrient mixture F12 (DMEM/F12 Gibco/Invitrogen) and supplemented with 10% fetal bovine serum (Biochrom). Whole-cell voltage clamp was performed with an EPC10 USB HEKA amplifier (HEKA Electronik). Signals were low passed at 1 kHz and sampled at 2–10 kHz. Patch pipettes were pulled from borosilicate glass tubes (TW150F-3; World Precision Instruments) to give a resistance of about 3 MΩ. Cells were held at −60 mV and all recordings were performed at room temperature. The external solution contained: 140 mM NaCl, 5 mM KCl, 2 mM MgCl$_2$, 5 mM EGTA, 10 mM glucose, and 10 mM HEPES, pH 7.4 (adjusted with NaOH). Calcium was omitted in order to avoid desensitization. The standard pipette solution contained: 140 mM KCl, 2 mM MgCl$_2$, 5 mM EGTA, and 10 mM HEPES, pH 7.4 (adjusted with KOH). A gravity-driven glass multi-barrel perfusion system was used to bath apply solutions. Data acquisition and off-line analyses required Patchmaster/Fitmaster software (HEKA Electronik) and Origin 8.5.1 (Origin Lab).

## TRPV5 electrophysiology

HEK293 cells (ATCC CRL-1573, RRID:CVVL_0045) maintained in MEM with 10% FBS and 1% penicillin/streptomycin at 37 °C with 5% CO$_2$ were transiently transfected using the Effectene reagent (Qiagen) according to the manufacturer's instructions with rbTRPV5-IRES (pCI-Neo vector) and its mutants. Two mutants, T539A and T539A/D542A, were generated with site-directed mutagenesis by BIONICS (Seoul, South Korea) and were confirmed by sequencing. Measurements with these mutants were performed in HEK293T cells, which were cultured in DMEM + 10% FBS. After 24 h, transfected cells were plated onto poly-D-lysine-coated 12-mm round coverslip (Fisher Scientific) and incubated for 24 h before whole-cell patch-clamp recordings. The currents were

measured using an Axopatch 200B amplifier (Molecular Devices) with a voltage ramp protocol from −80 mV to +80 mV. Currents were filtered at 2 kHz using a low-pass Bessel filter of the amplifier and digitized using a Digidata 1440 unit (Molecular Devices). Patch pipettes were pulled from borosilicate glass capillaries (O.D. 1.5 mm, I.D. 0.75 mm) (Sutter Instruments) of 4–6 MΩ resistance on a P-97 and P-1000 pipette puller (Sutter Instrument) and filled with a solution containing 140 mM K-gluconate, 10 mM HEPES, 5 mM EGTA, 2 mM $MgCl_2$, and 2 mM $Na_2ATP$ (pH = 7.3). The extracellular solution contained 142 mM LiCl, 1 mM $MgCl_2$, 10 mM HEPES, 10 mM glucose (pH = 7.4). In this solution, $Mg^{2+}$, and trace amounts of $Ca^{2+}$ block TRPV5 currents. Particularly, the D542A mutant is not $Ca^{2+}$ permeable, but conducts monovalent currents (Nilius et al, 2001c). Monovalent TRPV5 currents were initiated by the same solution containing 1 mM EGTA but no $MgCl_2$. LiCl in the extracellular solution was replaced with a equimolar larger cation NMDG-Cl to fully inhibit TRPV5 ion permeability. Different concentrations of RR (Calbiochem, cat. # 557450 or TOCRIS, catalog #1439) were applied to the cells to inhibit TRPV5 activation using a gravity-driven perfusion system.

## Data availability

The atomic coordinates and cryo-EM density maps of the structures presented in this paper are deposited in the Protein Data Bank and Electron Microscopy Data Bank under the accession codes TRPV2$_{RR}$ [PDB 8FFL and EMD-29046], TRPV2$_{2APB+RR}$ [PDB 8FFM and EMD-29047], TRPV5$_{RR}$ [PDB 8FFN and EMD-29048], and TRPV5 $_{PIP2+RR}$ [PDB 8FFQ and EMD-29051].

## Peer review

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

## Acknowledgements

We acknowledge the use of instruments at the Electron Microscopy Resource Lab and at the Beckman Center for Cryo-Electron Microscopy at the University of Pennsylvania Perelman School of Medicine. We also thank Stefan Steimle for assistance with Krios microscope operation at the University of Pennsylvania. We thank Sabine Baxter for assistance with hybridoma and cell culture at the University of Pennsylvania Perelman School of Medicine Cell Center Services Facility. This work was supported by a grants from the National Institute of Health (R35GM144120 to VM-B and R01GM093290 to TR).

## Author contributions

**Ruth A Pumroy**: Conceptualization; Formal analysis; Validation; Investigation; Visualization; Methodology; Writing—original draft; Writing—review and editing. **José J De Jesús-Pérez**: Conceptualization; Formal analysis; Validation; Investigation; Visualization; Methodology; Writing—original draft; Writing—review and editing. **Anna D Protopopova**: Conceptualization; Formal analysis; Validation; Investigation; Visualization; Methodology; Writing—original draft; Writing—review and editing. **Julia A Rocereta**: Formal analysis; Validation; Investigation; Methodology. **Edwin C Fluck**: Formal analysis; Validation; Investigation; Methodology. **Tabea Fricke**: Formal analysis; Validation; Investigation; Visualization; Methodology. **Bo-Hyun Lee**: Formal analysis; Validation; Investigation; Visualization; Methodology. **Tibor Rohacs**: Supervision; Funding acquisition; Writing—review and editing. **Andreas Leffler**: Formal analysis; Supervision; Validation; Investigation; Visualization; Methodology; Writing—review and editing. **Vera Moiseenkova-Bell**: Conceptualization; Supervision; Funding acquisition; Writing—original draft; Writing—review and editing.

## Disclosure and competing interests statement

The authors declare no competing interests.

# Expanded View Figures

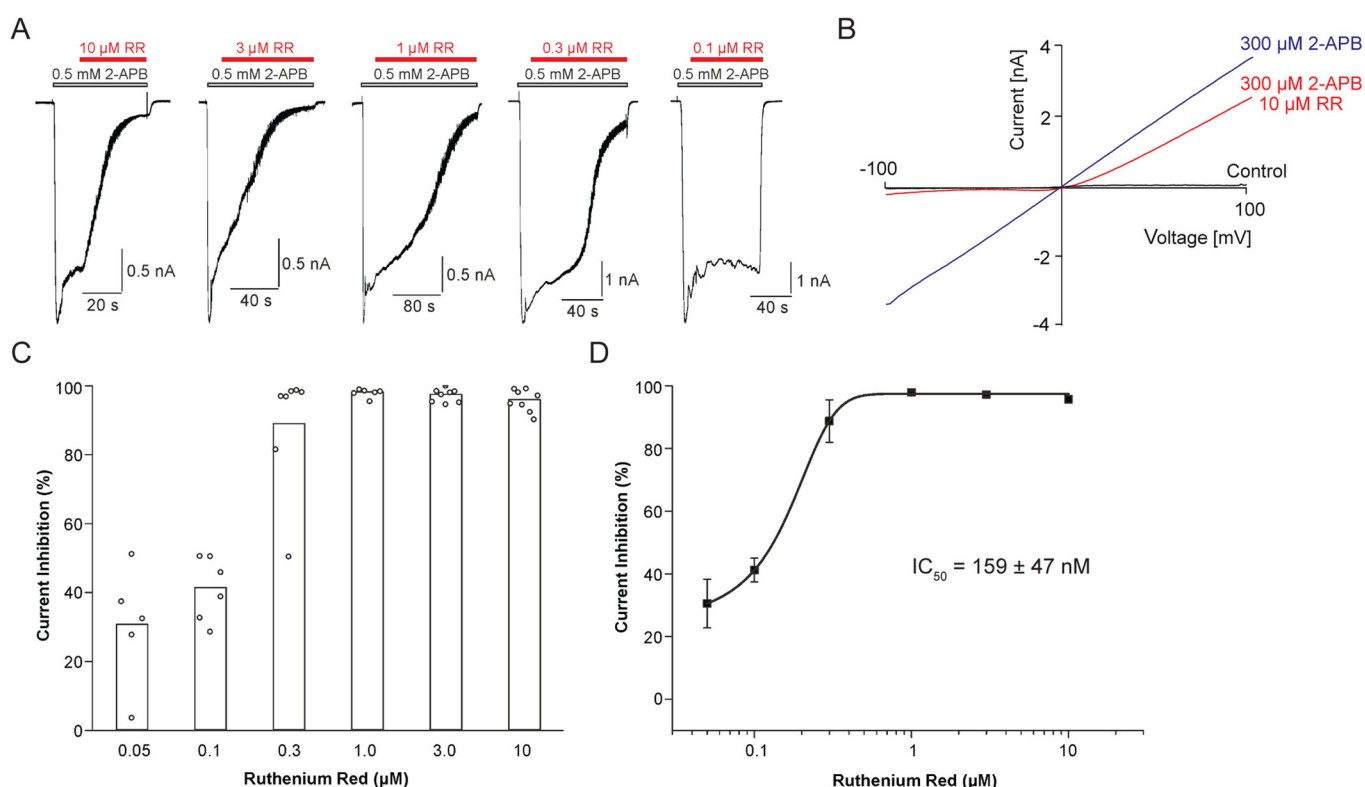

**Figure EV1.  Inhibition of 2-APB-induced TRPV2 currents by ruthenium red.**

(A) Representative current traces from HEK293 cells displaying inhibition of 2-APB-induced currents by RR at different concentrations. RR was co-applied with 0.5 mM 2-APB once the current induced by 2-APB applied alone had reached steady state. Cells were held at −60 mV, and only one concentration of RR was tested on each cell examined. (B) Typical current trace on rTRPV2 recording during a 500 ms long voltage-ramp ranging from −100 to 100 mV. Note that RR-induced inhibition of 2-APB-induced currents was more effective on inward currents than on outward currents. (C) Bar columns displaying the average inhibition of inward currents evoked by 2-APB for each examined concentration of RR. ([RR] in nM, number of cells for each concentration): (0, 8), (50, 5), (100, 6), (300, 7), (1000, 6), (3000, 8) and (10,000, 8). (D) Dose-response curve for RR-induced inhibition of inward currents evoked by 2-APB on rTRPV2. Mean ± S.E.M. fractional block induced by each concentration of RR are given. The data were fitted with the Hill1 equation with the Origin software. Cells for each concentration are the same as (C): ([RR] in nM, number of cells for each concentration): (0, 8), (50, 5), (100, 6), (300, 7), (1000, 6), (3000, 8) and (10,000, 8). Source data are available online for this figure.

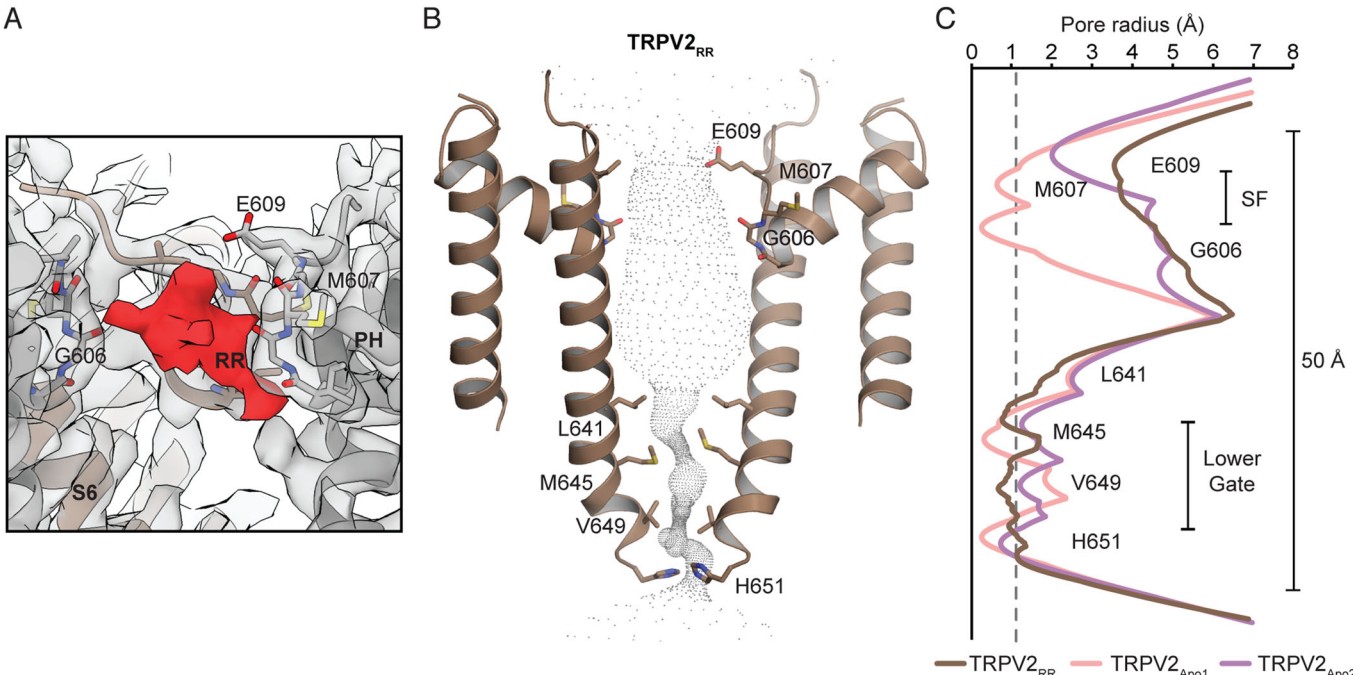

**Figure EV2. Selectivity filter and pore of TRPV2$_{RR}$.**

(A) Map density and cartoon representation of the possible RR binding site in TRPV2$_{RR}$. One monomer is colored brown, the adjacent monomers are colored gray. Putative density for RR is colored red. The map is contoured at σ = 5. (B) HOLE-generated pore profile of TRPV2$_{RR}$. (C) Graphical representation of the pore profiles of TRPV2$_{RR}$ (brown), TRPV2$_{Apo1}$ (PDB 6U84, pink) and TRPV2$_{Apo2}$ (PDB 6U86, purple). The radius of a dehydrated calcium ion is marked by a dotted gray line at 1.1 Å.

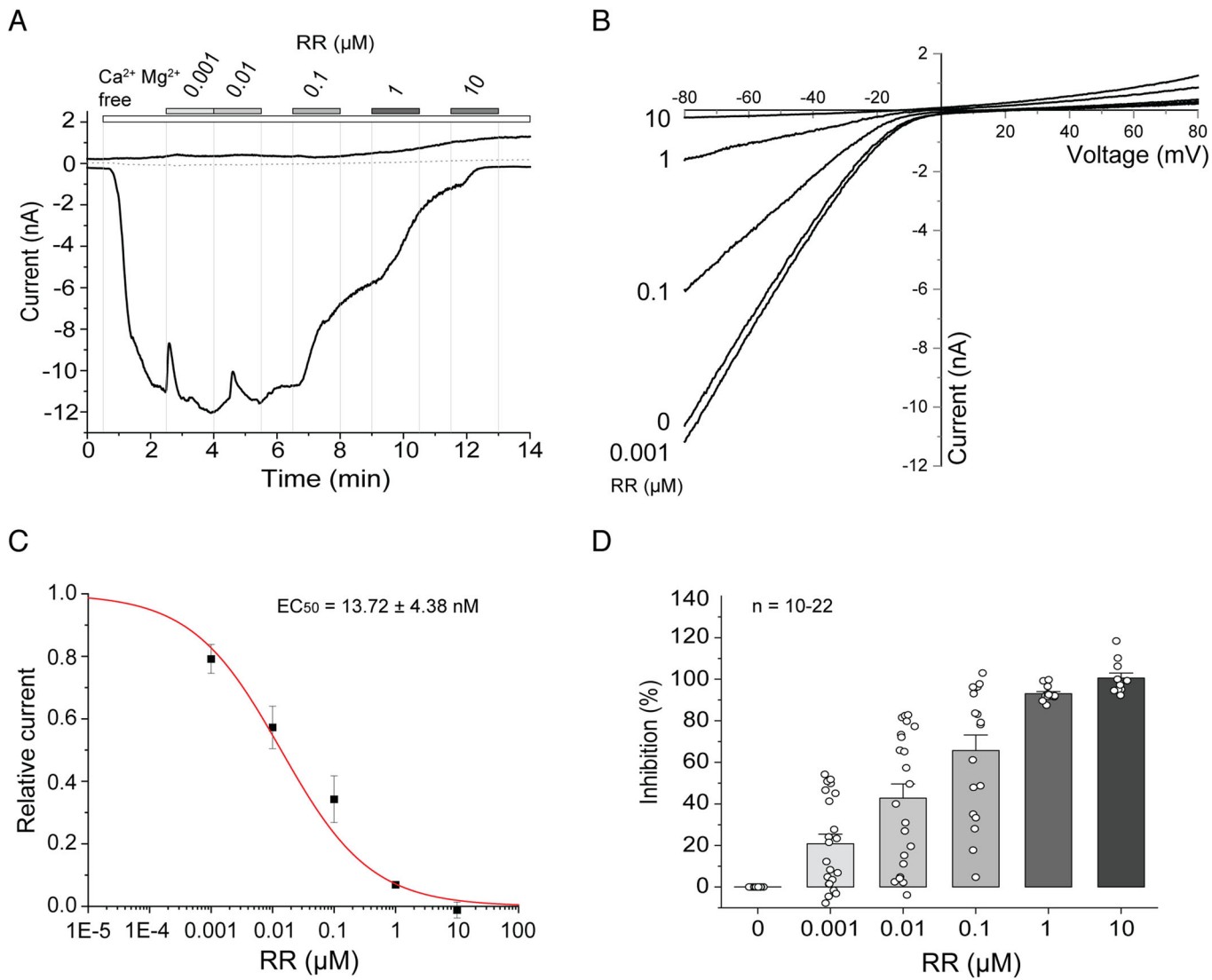

**Figure EV3. Inhibition of TRPV5 currents by Ruthenium red.**

(A) Representative current traces from HEK293 cells transfected with TRPV5, at 80 mV (upper trace) and −80 mV (lower trace). Monovalent currents were initiated by the application of a $Ca^{2+}$ and $Mg^{2+}$ free solution, as described in the methods section. Dotted line shows zero current, and applications of various concentrations of RR are shown with horizontal bars. (B) Representative ramp current–voltage (I–V) traces from (A) revealed characteristic inwardly rectifying TRPV5 currents in the absence or presence of different concentrations of RR. (C) Relative current levels at −80 mV after application of the various concentrations of RR. The data were fitted using the Hill1 equation with the Origin2021 software and plotted as mean ± SEM. ([RR] in nM, number of cells for each concentration): (0, 22), (1, 22), (10, 22), (100, 18), (1000, 12), and (10,000, 10). (D) Bar graph (mean ± SEM and individual values) of inhibition of TRPV5-mediated monovalent currents by various concentrations of RR. Cells for each concentration are the same as (C): ([RR] in nM, number of cells for each concentration): (0, 22), (1, 22), (10, 22), (100, 18), (1000, 12), and (10,000, 10). Source data are available online for this figure.

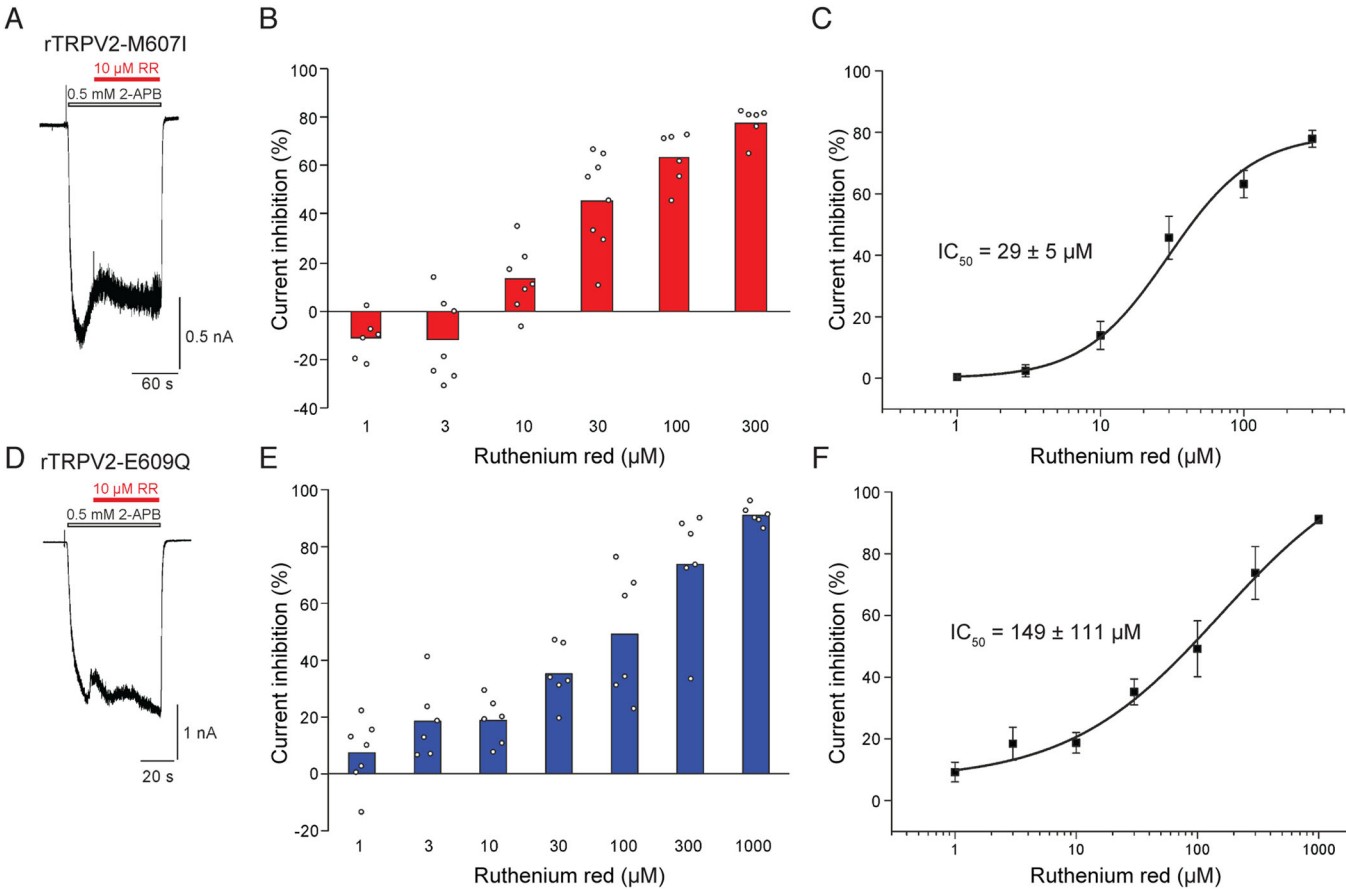

**Figure EV4. Mutations to the TRPV2 selectivity filter reduce RR pore block.**

(A) Representative current trace displaying inhibition of 2-APB-induced current by 10 μM RR on the mutant rTRPV2-M607I from HEK293 cells. (B) RR was co-applied with 0.5 mM 2-APB once the current induced by 2-APB applied alone had reached steady state. HEK293 Cells were held at −60 mV. (B, E) Bar columns displaying the average inhibition of inward currents evoked by 2-APB for each examined concentration of RR on rTRPV2-M607I. ([RR] in μM, number of cells for each concentration): (0, 8), (1, 6), (3, 7), (10, 7), (30, 8), (100, 6) and (300, 6). (C) Dose–response curve (mean ± SEM) for RR-induced inhibition of inward currents evoked by 2-APB on rTRPV2-M607I. Cells for each concentration are the same as (B): ([RR] in μM, number of cells for each concentration): (0, 8), (1, 6), (3, 7), (10, 7), (30, 8), (100, 6) and (300, 6). (D) Representative current trace displaying inhibition of 2-APB-induced currents by 10 μM RR on the mutant rTRPV2-E609Q. (E) RR was co-applied with 0.5 mM 2-APB once the current induced by 2-APB applied alone had reached steady state. HEK293 Cells were held at −60 mV. (B, E) Bar columns displaying the average inhibition of inward currents evoked by 2-APB for each examined concentration of RR on rTRPV2-E609Q. $n = 6$–8 cells for each concentration. ([RR] in μM, number of cells for each concentration): (0, 7), (1, 7), (3, 6), (10, 6), (30, 6), (100, 6), (300, 6), and (1000, 6). (F) Dose–response curves for RR-induced inhibition of inward currents evoked by 2-APB on rTRPV2-E609Q. Mean ± S.E.M. fractional block induced by each concentration of RR are given. The data were fitted with the Hill1 equation with the Origin software. Cells for each concentration are the same as (F): ([RR] in μM, number of cells for each concentration): (0, 7), (1, 7), (3, 6), (10, 6), (30, 6), (100, 6), (300, 6), and (1000, 6). Source data are available online for this figure.

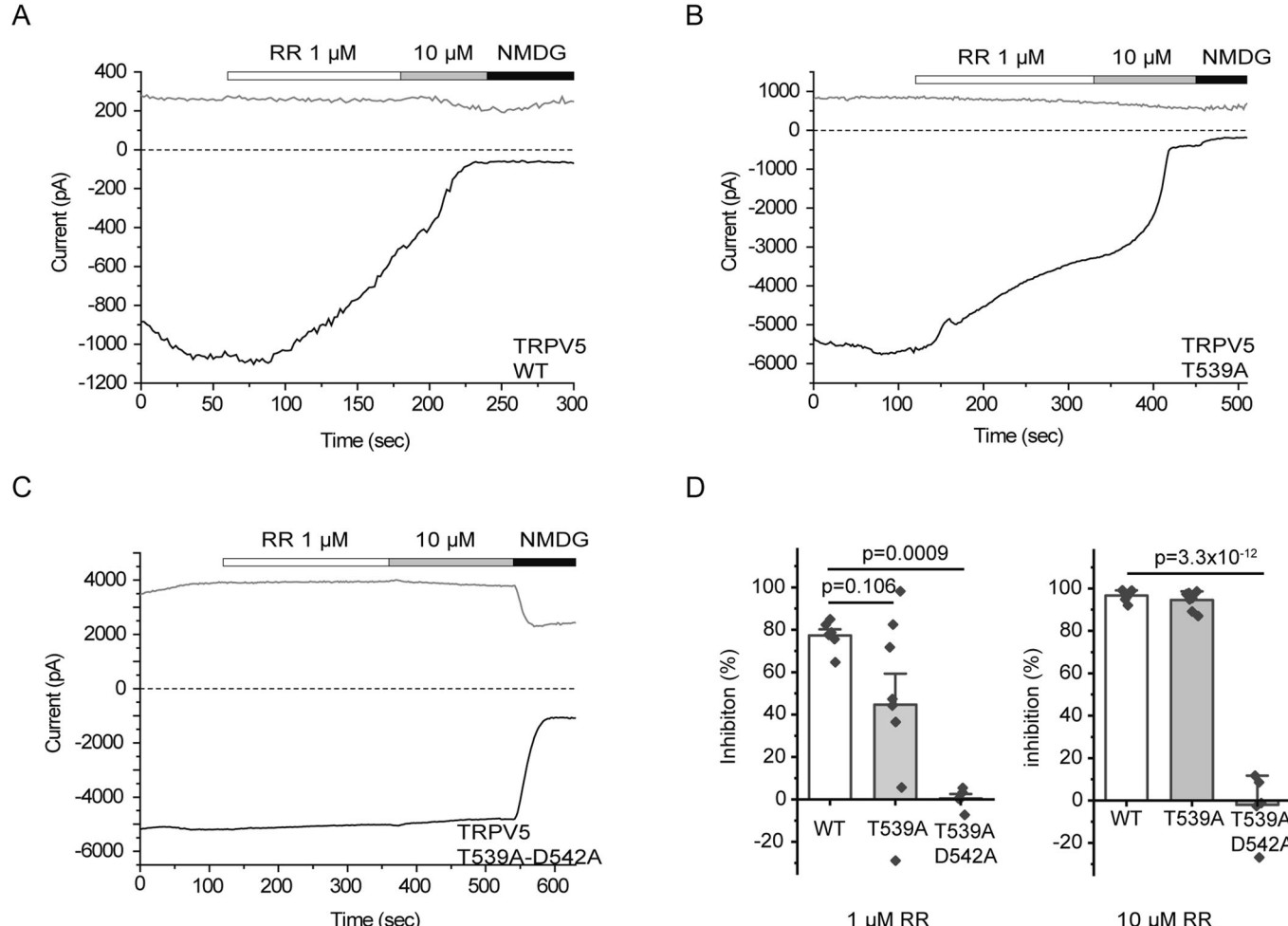

**Figure EV5.  Mutations to the TRPV5 selectivity filter reduce RR pore block.**

(A–C) Representative trace for TRPV5 WT (A), T539A (B), and T539A/D542A (C) currents at 80 mV (upper trace) and −80 mV (lower trace). The application of a $Ca^{2+}$ and $Mg^{2+}$ free solution was used as a bath solution (extracellular solution) to initiate TRPV5 currents. The dotted line shows zero current, and applications of 1 or 10 μM of RR are shown with horizontal bars. At the last minute of recording, NMDG was applied to inhibit ion permeability through the TRPV5 pore. (D) Bar graph (mean ± SEM and individual values) of inhibition of WT or mutant TRPV5 currents by 1 or 10 μM RR. $n = 6$ cells transfected with WT TRPV5 for 1 or 10 μM RR, $n = 8$ for T539A, and $n = 5$ for T539A + D542A. Statistical significance was calculated with one-way analysis of variance, with Bonferroni's post hoc test. Source data are available online for this figure.

