## [Peer Review File · EMBO Reports]

Molecular Details of Ruthenium Red Pore Block in TRPV Channels

Vera Moiseenkova-Bell, Tibor Rohacs, Andreas Leffler, Ruth Pumroy, José De Jesús-Pérez, Anna Protopopova, Julia Rocereta, Edwin Fluck, Tabea Fricke, and Bo Lee

DOI: [10.15252/embr.202358051](https://doi.org/10.15252/embr.202358051)

Corresponding author: Vera Moiseenkova-Bell (vmb@pennmedicine.upenn.edu)

Review Timeline:

Submission Date:	24th Aug 23
Editorial Decision:	16th Oct 23
Revision Received:	28th Nov 23
Editorial Decision:	19th Dec 23
Revision Received:	19th Dec 23
Accepted:	21st Dec 23

Transaction Report:

Dear Prof. Moiseenkova-Bell

Thank you for the submission of your research manuscript to our journal. We have now received the full set of referee reports that is copied below.

Given the supportive and constructive comments, we would like to invite you to revise your manuscript with the understanding that the referee concerns must be fully addressed and their suggestions taken on board. Please address all referee concerns in a complete point-by-point response. Acceptance of the manuscript will depend on a positive outcome of a second round of review. It is EMBO Reports policy to allow a single round of revision only and acceptance or rejection of the manuscript will therefore depend on the completeness of your responses included in the next, final version of the manuscript.

We realize that it is difficult to revise to a specific deadline. In the interest of protecting the conceptual advance provided by the work, we recommend a revision within 3 months (January 16, 2023). Please discuss the revision progress ahead of this time with the editor if you require more time to complete the revisions.

I am also happy to discuss the revision further via e-mail or a video call, if you wish.

*****IMPORTANT NOTE:

We perform an initial quality control of all revised manuscripts before re-review. Your manuscript will FAIL this control and the handling will be delayed IN CASE the following APPLIES:

- 1) A data availability section providing access to data deposited in public databases is missing. If you have not deposited any data, please add a sentence to the data availability section that explains that.
- 2) Your manuscript contains statistics and error bars based on $n=2$. Please use scatter blots in these cases. No statistics should be calculated if $n=2$.

When submitting your revised manuscript, please carefully review the instructions that follow below. Failure to include requested items will delay the evaluation of your revision.*****

- 2) individual production quality figure files as .eps, .tif, .jpg (one file per figure). Please download our Figure Preparation Guidelines (figure preparation pdf) from our Author Guidelines pages <https://www.embopress.org/page/journal/14693178/authorguide> for more info on how to prepare your figures.

- 4) a complete author checklist, which you can download from our author guidelines (<<https://www.embopress.org/page/journal/14693178/authorguide>>). Please insert information in the checklist that is also reflected in the manuscript. The completed author checklist will also be part of the RPF.

- 5) Please note that all corresponding authors are required to supply an ORCID ID for their name upon submission of a revised manuscript (<<https://orcid.org/>>). Please find instructions on how to link your ORCID ID to your account in our manuscript tracking system in our Author guidelines (<<https://www.embopress.org/page/journal/14693178/authorguide#authorshipguidelines>>)

- 6) We replaced Supplementary Information with Expanded View (EV) Figures and Tables that are collapsible/expandable online.

A maximum of 5 EV Figures can be typeset. EV Figures should be cited as 'Figure EV1, Figure EV2' etc... in the text and their respective legends should be included in the main text after the legends of regular figures.

7) The Data Availability section should follow the model below (see also <<https://www.embopress.org/page/journal/14693178/authorguide#dataavailability>>).

Data availability

Additional information on source data and instruction on how to label the files are available <<https://www.embopress.org/page/journal/14693178/authorguide#sourcedata>>.

10) Figure legends and data quantification:

- the name of the statistical test used to generate error bars and P values,
- the number (n) of independent experiments (please specify technical or biological replicates) underlying each data point,
- the nature of the bars and error bars (s.d., s.e.m.)
- If the data are obtained from n {less than or equal to} 5, show the individual data points in addition to the SD or SEM.
- If the data are obtained from n {less than or equal to} 2, use scatter blots showing the individual data points.

11) Our journal encourages inclusion of *data citations in the reference list* to directly cite datasets that were re-used and obtained from public databases. Data citations in the article text are distinct from normal bibliographical citations and should directly link to the database records from which the data can be accessed. In the main text, data citations are formatted as follows: "Data ref: Smith et al, 2001" or "Data ref: NCBI Sequence Read Archive PRJNA342805, 2017". In the Reference list, data citations must be labeled with "[DATASET]". A data reference must provide the database name, accession number/identifiers and a resolvable link to the landing page from which the data can be accessed at the end of the reference. Further instructions are available at <<https://www.embopress.org/page/journal/14693178/authorguide#referencesformat>>.

12) All Materials and Methods need to be described in the main text. We would encourage you to use 'Structured Methods', our new Materials and Methods format. According to this format, the Materials and Methods section should include a Reagents and Tools Table (listing key reagents, experimental models, software and relevant equipment and including their sources and

relevant identifiers) followed by a Methods and Protocols section in which we encourage the authors to describe their methods using a step-by-step protocol format with bullet points, to facilitate the adoption of the methodologies across labs.

More information on how to adhere to this format as well as downloadable templates (.doc or .xls) for the Reagents and Tools Table can be found in our author guidelines: <

<https://www.embopress.org/page/journal/14693178/authorguide#manuscriptpreparation>>.

<<https://www.embopress.org/doi/10.15252/msb.20178071>>.

13) As part of the EMBO publication's Transparent Editorial Process, EMBO Reports publishes online a Review Process File to accompany accepted manuscripts. This File will be published in conjunction with your paper and will include the referee reports, your point-by-point response and all pertinent correspondence relating to the manuscript.

Yours sincerely,

Referee #1:

The manuscript by Pumroy et al., describes the molecular detail of ruthenium red (RR) on pore blocking in TRPV channels. Overall the study is of a high quality and the cryoEM structures are clear. The study will be important in the field and the authors should be commended for their work. There were no major concerns with the work, however there are a few minor comments that may increase the clarity and/or impact of the study (in order of appearance).

The introduction is comprehensive to anyone who knows TRPV biochemistry but could do with a couple details for those who are not, particularly a few words briefly explaining a knock on/knock off mechanism would be appreciated.

For consistency some units have a space before others don't for example 1mM vs 0.5 mM

RR was added to ensure "saturation" it would be good to give what the concentration required would be without voltage, ie not through patch clamp? Has this been studied previously or is it unknown? Does this then link to the 2 potential binding positions, is one an artifact of the high concentration of RR?

For the datasets, its interesting that the PIP2 dataset contains a significant amount more particles per micrograph almost by an order of magnitude. Why is this? Given the significant amount of data here, why only a small change in resolution.

It is interesting that when PIP2 is added the sample behaves very differently in terms of preferred orientation and the micrograph shown. Do the authors know why this might be?

When comparing conformations it would be good to see an r.m.s.d value, maybe a table comparing these would be a good way to display this.

For the methods, line 2, at 300c with 200. 200 what? Assume r.p.m.

What wavelength was used for the O.D.

How was the sample concentrated?

What temperature were the centrifugal steps undertaken at, one would assume 4 Deg?

What size Superose 6 column was used?

Second paragraph, third line, should be HCl not HCL.

What were the glow discharge parameters used?

For the TRPV5RR dataset why was this the only one which used CRYOSPARC.

In Supp table 1, why does the pixel size vary from 0.834, 0.84, 0.83. In addition the dose should also be given, B-factors for the model would help.

Labels in Supp 2 A and D would help identify the map seen as is done in supp fig 8.

Figure 1B, the density for 2-APB is not clear and in the current view it looks to be a poor fit.

Referee #2:

This manuscript from Pumroy etc. report the cryo-EM structures of TRPV2 and TRPV5 in complex with Ruthenium red, which is a channel blocker, in presence and absence of agonist and then validate these structural finding using electrophysiology. Transient Receptor Potential (TRP) channels, which are generally non-selective cation channels, conduct Ca²⁺ in response to disparate activators, including sensory stimuli such as temperature, touch, and pungent chemicals. This manuscript is important for the TRP channel community because it provides a RR based structural-functional framework for better understanding TRP channel block. Although the structure of TRPV6, a homolog of TRPV5 and TRPV2, complexed with RR has been determined, the structures presented here provide new additional insight into regulation of channel activity. However, however, there are some important criticisms that we would first like to be addressed.

1. Authors mentioned that "As this density is not present in the otherwise similar TRPV2Apo2 (PDB 6U86/EMD-20678) structure, we attribute this density to RR, although the irregular shape does not allow for the molecule to be built. This irregular shape may be due to the flexibility of the TRPV2 SF, giving RR the freedom to sample multiple binding conformations" but this in contrast to the structure solved in presence of 2-ABP, where the RR has bound to the channel with explicitly RR, snugly fitting in the predicted density. Therefore, RR in the TRPV2 structure where 2-APB is not present need further clarification. It would be interesting to see whether the focused refinement or signal subtraction improves the density for RR in TRPV2 SF.
2. Supplementary figures 3 indicate the use of C4 symmetry in the TRPV2-2APB+RR structure and C1 symmetry in the TRPV2+RR structure. Is there a specific reason for the symmetry change in the TRPV2+RR structure? Is the pore still exhibiting 4-fold symmetry?
3. The sentence "the addition of RR to the open channel led to the pore reverting to the closed conformation" does not clarify whether RR also binds to the closed state and stabilize it. It is possible that RR stabilizes the closed state. Considering that PIP2 only binds to a small fraction of particles, it is also possible that RR does not stabilize the PIP2-bound open state rather inhibited state since PIP2 is not present in the structure. The authors should exercise caution in their analysis regarding these possibilities.
4. The sentence "RR is not large enough to simultaneously occupy all three positions" authors need to explicitly state which three position are these?
5. The authors suggest that the irregular shape of RR density in TRPV2-RR may be due to the flexibility of the TRPV2 SF, allowing RR to adopt various binding conformations. However, in the presence of 2-APB, the RR density is well-defined and matches the predicted density. Further clarification is needed to clarify the RR density in TRPV2 when 2-APB is absent. It would be valuable to explore focused refinement or signal subtraction to improve RR density in the TRPV2 SF and see if the predicted density matches with that of RR.
6. The text mentions that "it is unclear if RR binding in the TRPV2RR SF represents a blocked state of the channel." It would be helpful to elaborate on the potential consequences or significance of this uncertainty. What are the broader implications of this observation for the ion channel field?

7. When referencing previous work, it is important to include citations to support the claims made. For instance, when mentioning the "TRPV2Apo2 state (PDB 6U86)," provide a citation to the relevant paper or publication.
8. Previously, it was established that substituting D542A in the SF of TRPV5 or TRPV6 renders the channel nonfunctional. Therefore, it's essential to clarify how currents are obtained in a double mutation (T539A/D542A) channel and how RR inhibits this channel.
9. What causes the difference in RR binding when PIP2 is present compared to when it is absent? These structural findings are very interesting and needs to be explained in the text.
10. Despite RR having a higher affinity for the channel, TRPV2-2APB+RR still appears to be in an open-like state, rather than inhibited or closed. It is worth investigating whether any other channel conformations were excluded during data processing, possibly representing an inactivated state. Additionally, why isn't PIP2 observed in the TRPV5 structure? It is possible that PIP2-containing particles were excluded from data processing because they represent only a subset of particles. The data processing flow chart shows many classes that could be refined to a decent resolution, making it interesting to explore these different conformations.
11. Possibly beyond the scope of this article, but I am very curious how your data set might benefit from 3D variability analysis and/or multi-body refinement for the two particle stacks individually and/or also combined? It would be great to see if undulations coincide with RR bindings, and/or improvements in resolutions using multi-body refinement approaches
12. How the lipids are placed in the channel in these structures and compared to previously published structures? Do authors observe any differences in lipid movement or displacement in RR bound state? I think a figure is required for comparing the lipid presence or absence or displacement in different conformation of these channels.

Referee #3:

The manuscript entitled "Molecular Details of Ruthenium Red Pore Block in TRPV Channels" submitted by Pumroy et al and by Vera Moiseenkova-Bell lab is an interesting manuscript.

Ruthenium Red has been used as a channel blocker for a long time. In this manuscript, the authors attempted to understand the binding site of the Ruthenium Red on TRPV2 and TRPV5, a thermosensitive and a non-thermosensitive ion channel respectively. Though this work is good and a lot of work has been included in this manuscript, at present this manuscript lacks some key aspects. My comments are not the quality of the data that has been provided. My comments are mainly to address the questions and confirming the findings.

Major concerns are:

- 1) No mutagenesis work has been done to confirm the role of main residues that confer Ruthenium Red binding. Ideally these few key residues should have been mutated and electrophysiology data should be provided to show that efficacy of Ruthenium Red binding on the mutated channel. In absence of this mutagenesis followed by electrophysiology data
- 2) Simulation data is required to supplement the binding of Ruthenium Red. The simulation data will provide more mechanistic details.
- 3) Another major concern is the introduction section. The logic to pick up only TRPV2 and TRPV5 among all other TRP channels are not properly justified.
- 4) Also use of 2APB as TRPV2 agonist is bit critical. 2APB is not a specific ligand for TRPV2 and it binds to many other ion channels. In that context also logic to use 2APB as a ligand for TRPV2 need justification. The point is there might be a different structure altogether if the ligand become highly-specific for a channel.
- 5) Is there any mutation known in the human disease data base/cancer data base that high-lights the key residues involved in Ruthenium Red binding on the TRPV2 and TRPV5? Especially in the pore region?

Referee #1:

The manuscript by Pumroy et al., describes the molecular detail of ruthenium red (RR) on pore blocking in TRPV channels. Overall the study is of a high quality and the cryoEM structures are clear. The study will be important in the field and the authors should be commended for their work. There were no major concerns with the work, however there are a few minor comments that may increase the clarity and/or impact of the study (in order of appearance).

We thank the reviewer for their thoughtful comments.

The introduction is comprehensive to anyone who knows TRPV biochemistry but could do with a couple details for those who are not, particularly a few words briefly explaining a knock on/knock off mechanism would be appreciated.

We thank the reviewer for pointing this out, we have added a brief explanation of the knock off mechanism to the introduction. We added the following sentence to the first paragraph of the introduction:

“In this mechanism, the SF can host multiple ions simultaneously, then when a new Ca^{2+} ion is introduced into SF, it displaces or “knocks off” the originally positioned Ca^{2+} ion from the pore, making room for the incoming one”

For consistency some units have a space before others don't for example 1mM vs 0.5 mM

We thank the reviewer for catching this, we have gone over the text and figures carefully to make sure this formatting is consistent.

RR was added to ensure "saturation" it would be good to give what the concentration required would be without voltage, ie not through patch clamp? Has this been studied previously or is it unknown? Does this then link to the 2 potential binding positions, is one an artifact of the high concentration of RR?

We are not aware of any non-patch clamp experiments, such as cytoplasmic Ca^{2+} measurements examining the effect of RR on TRPV5 and TRPV2, but keep in mind that the membrane potential of most cells is negative, so even in those experiments the membrane potential would not be zero. In patch clamp experiments it is hard to determine RR block at 0 mV, as the currents reverse there. As for TRPV5, due to strong inward rectification, it is also hard to determine RR block at positive voltages. Given that TRPV2 is inhibited less at positive voltages, we may assume that inhibition also requires more RR at 0 mV than at negative voltage. For this reason, and to align with established practices in cryo-EM structural studies of ion channels [Pope et al., 2020 (PMCID: PMC7245552); Neuberger et al., 2021 (PMCID: PMC8560856); Zhen et al., 2023 (PMCID: PMC10333285)], we employed 1 mM RR.

With regard to the 2 potential binding positions, that result could be due to the different initial conformational states. When RR is closer to D542 in TRPV5, it has not been subjected to any activation stimulus. Conversely, when RR is positioned nearer to N572, the channel was previously activated with PIP2.

For the datasets, its interesting that the PIP2 dataset contains a significant amount more particles per micrograph almost by an order of magnitude. Why is this? Given the significant amount of data here, why only a small change in resolution.

It can be challenging to make a clear connection between grid quality and freezing conditions as samples that are apparently identical can result in grids of widely varying quality. That being said, we have observed that the addition of charged lipids like PIP2 can have a big effect on particle distribution, perhaps due to some effect at the air-water interface.

With regards to the small change in resolution despite a large difference in the number of particles, there are a variety of factors that can contribute to the final structure quality beyond particle count, such as ice thickness, heterogeneity and angular distribution. In this case, it's important to consider that the activator directly influences the conformational state of the channel. In the TRPV5 RR and PIP2+RR datasets, the RR dataset has very little heterogeneity compared to the increased heterogeneity of the PIP2+RR dataset.

It is also worth noting that the highest resolution structure of TRPV5 currently deposited in the RCSB database is 2.6 Å, very close to the 2.65 Å we report here for TRPV5_{PIP2+RR}. Additionally, the difference in resolution between the TRPV5_{RR} and TRPV5_{PIP2+RR} structures is 0.3 Å, which is a significant improvement at this resolution range.

It is interesting that when PIP2 is added the sample behaves very differently in terms of preferred orientation and the micrograph shown. Do the authors know why this might be?

We have observed that the addition of PIP2 can have an effect on particle distribution and grid quality compared to freezing apo conditions of the same batch of protein on the same day, but the effects are not consistent enough to establish a clear pattern. As we show in Appendix Figure S5 (initially called Supplementary Figure 8), there's a large change in angular distribution among the three conditions with RR and PIP2. Difference in particle behavior may be due to the addition of charge or perhaps due to some change at the air-water interface.

When comparing conformations it would be good to see an r.m.s.d value, maybe a table comparing these would be a good way to display this.

We thank the reviewer for the suggestion, we have added r.m.s.d. values in the main text when comparing the structures presented in this paper to previously published structures.

For the methods, line 2, at 300c with 200. 200 what? Assume r.p.m.

What wavelength was used for the O.D.

How was the sample concentrated?

What temperature were the centrifugal steps undertaken at, one would assume 4 Deg?

What size Superose 6 column was used?

Second paragraph, third line, should be HCl not HCL.

What were the glow discharge parameters used?

We have added these details to the methods section

For the TRPV5RR dataset why was this the only one which used CRYOSPARC.

We have only included the processing trees done to obtain the final published structures, but we did extensive analysis in both cryoSPARC and Relion to determine what states were contained in each dataset and the processing strategy to get the best structures. We tended to favor Relion over cryoSPARC for processing due to the Bayesian polishing feature, but did move particles between programs for certain features. For example, the TRPV2_{2APB+RR} dataset was primarily processed in Relion, but the final set of particles were imported to cryoSPARC for non-uniform refinement as a final step.

In Supp table 1, why does the pixel size vary from 0.834, 0.84, 0.83. In addition the dose should also be given, B-factors for the model would help.

While all of the data was collected on the same microscope, the collections were separated by several months. This resulted in some slight variations in pixel size at the same magnification, especially as we had to swap out the camera during this period.

We have added the dose and model B-factors to Appendix Table S1 (before called Supplementary Table 1).

Labels in Supp 2 A and D would help identify the map seen as is done in supp fig 8.

We thank the reviewer for this comment and have made this alteration to Appendix Figure S1 (initially called Supplemental Figure 2).

Figure 1B, the density for 2-APB is not clear and in the current view it looks to be a poor fit.

We agree that the density for 2-APB is not ideal. This is due to the combined factors of poor affinity of 2-APB (EC_{50} in the hundreds of micromolar range) for TRPV2 and the heterogeneity of TRPV2 induced by 2-APB binding. While the density for 2-APB is not as clear as the density we observe for RR, it is entirely consistent with both our previously observed 2-APB density (Pumroy et al Nat Comm 2022) and the conformational changes we observe at S5 and the S4-S5 linker which create a gap at this site, as we mention in the text. Beyond the structures the reviewer has access to (in this work and our previous publications), I should note that we have obtained many structures of TRPV2 in the presence of 2-APB in a variety of different conditions and 2-APB produces a strong and incredibly consistent structural effect on TRPV2 in our hands.

Referee #2:

This manuscript from Pumroy etc. report the cryo-EM structures of TRPV2 and TRPV5 in complex with Ruthenium red, which is a channel blocker, in presence and absence of agonist and then validate these structural finding using electrophysiology. Transient Receptor Potential (TRP) channels, which are generally non-selective cation channels, conduct Ca^{2+} in response to disparate activators, including sensory stimuli such as temperature, touch, and pungent chemicals. This manuscript is important for the TRP channel community because it provides a RR based structural-functional framework for better understanding TRP channel block. Although the structure of TRPV6, a homolog of TRPV5 and TRPV2, complexed with RR has been determined, the structures presented here provide new additional insight into regulation of channel activity. However, however, there are some important criticisms that we would first like to be addressed.

We thank the reviewer for their thoughtful comments

1. Authors mentioned that "As this density is not present in the otherwise similar TRPV2Apo2 (PDB 6U86/EMD-20678) structure, we attribute this density to RR, although the irregular shape does not allow for the molecule to be built. This irregular shape may be due to the flexibility of the TRPV2 SF, giving RR the freedom to sample multiple binding conformations" but this in

contrast to the structure solved in presence of 2-APB, where the RR has bound to the channel with explicitly RR, snugly fitting in the predicted density. Therefore, RR in the TRPV2 structure where 2-APB is not present need further clarification. It would be interesting to see whether the focused refinement or signal subtraction improves the density for RR in TRPV2 SF.

While the processing for the structure we chose to publish in this work is the result of a simple processing method, we did make extensive attempts to improve density/alignment of the potential RR density in multiple programs, including focused refinement. We suspect that RR can occupy a large number of conformations in the open SF that unfortunately do not seem to coincide with even minor conformational changes to the channel which would usually aid in this kind of classification.

2. Supplementary figures 3 indicate the use of C4 symmetry in the TRPV2-2APB+RR structure and C1 symmetry in the TRPV2+RR structure. Is there a specific reason for the symmetry change in the TRPV2+RR structure? Is the pore still exhibiting 4-fold symmetry?

We chose to present the TRPV2_{RR} structure in C1 rather than C4 due to the amorphous nature of the extra density in the SF. When C4 symmetry was applied, this averaged to a central vertical density which would position RR outside interaction range of any pore residues, which we consider an artifact (Appendix Figure S4). The channel itself does remain C4 symmetrical, with only some slight variation in the pose of Glu609.

3. The sentence "the addition of RR to the open channel led to the pore reverting to the closed conformation" does not clarify whether RR also binds to the closed state and stabilize it. It is possible that RR stabilizes the closed state. Considering that PIP2 only binds to a small fraction of particles, it is also possible that RR does not stabilize the PIP2-bound open state rather inhibited state since PIP2 is not present in the structure. The authors should exercise caution in their analysis regarding these possibilities

We forgot to mention in the initial submission of this paper that TRPV5 was incubated with PIP2 for 5-10 minutes before the addition of RR rather than the two drugs being co-applied. In our cryoEM datasets of PIP2 alone, the open conformation is a major state immediately obvious even before sorting. Based on this, we think RR is indeed closing PIP2 activated TRPV5. We have clarified grid preparation conditions in the results section and methods.

4. The sentence "RR is not large enough to simultaneously occupy all three positions" authors need to explicitly state which three position are these?

We thank the reviewer for catching this mistake, this section was rearranged after being initially written and previously came after our discussion of the three proposed Ca²⁺ binding sites described for TRPV6 that we show in Figure 3. We have altered this line to remove the reference to the Ca²⁺ binding positions.

5. The authors suggest that the irregular shape of RR density in TRPV2-RR may be due to the flexibility of the TRPV2 SF, allowing RR to adopt various binding conformations. However, in the presence of 2-APB, the RR density is well-defined and matches the predicted density. Further clarification is needed to clarify the RR density in TRPV2 when 2-APB is absent. It would be valuable to explore focused refinement or signal subtraction to improve RR density in the TRPV2 SF and see if the predicted density matches with that of RR.

As mentioned above, we did make extensive attempts to resolve the conformation of RR in TRPV2_{RR}. Unfortunately, RR is either too small, too poorly coordinated, or too heterogeneous to be resolved for that dataset.

6. The text mentions that "it is unclear if RR binding in the TRPV2_{RR} SF represents a blocked state of the channel." It would be helpful to elaborate on the potential consequences or significance of this uncertainty. What are the broader implications of this observation for the ion channel field?

We thank the reviewer for pointing out that this sentence touches on a much bigger implication for the ion channel field. We decided to remove this sentence because based on the data we have, all we can say is that the SF in TRPV1-TRPV4 is flexible even in the absence of activators, as shown previously both functionally and structurally [Jara-Oseguera et al eLife (PMID: 31724952) and Pumroy et al eLife (PMID: 31566564)] and that RR does not take on a stable conformation with this wide SF. Now the sentence reads:

"Thus, our results agree with the previous observation that the TRPV1-TRPV3 SF is dynamic and can permit cation passage even when the pore is closed at the lower gate."

7. When referencing previous work, it is important to include citations to support the claims made. For instance, when mentioning the "TRPV2_{Apo2} state (PDB 6U86)," provide a citation to the relevant paper or publication.

We thank the reviewer for catching this oversight, we have added the reference.

8. Previously, it was established that substituting D542A in the SF of TRPV5 or TRPV6 renders the channel nonfunctional. Therefore, it's essential to clarify how currents are obtained in a double mutation (T539A/D542A) channel and how RR inhibits this channel.

While the reviewer is correct that the D542A mutant of TRPV6 is non-functional, the same mutant in TRPV5 is not permeable to Ca²⁺, but it still conducts monovalent currents [Nilius et al Cell Ca 2001 (PMID: 11352507) and Nilius et al JBC 2001 (PMID: 11035011)]. We measured monovalent currents in our experiments. This is described in the revised MS in the results section.

9. What causes the difference in RR binding when PIP2 is present compared to when it is absent? These structural findings are very interesting and needs to be explained in the text.

It's not entirely clear to us why the RR position differs, but our findings suggest an intimate connection between RR positioning within the pore and channel state. Specifically, when we compare the pores of TRPV5_{Apo} and TRPV5_{PIP2}, we observe that the lower gate is wider in the activated state than in the apo state (as shown in Fig. 2H). This aligns with the notion that in the activated state, the passage of cations is facilitated. We conjecture that RR, in its attempt to traverse the pore like any other cation, encounters obstacles due to its physicochemical properties, notably its charge, and its size, and particularly due to the interaction with the N572. These characteristics collectively impede its movement within the permeation pathway, causing blockage of the channel and consequent closure. We have added this concept to the discussion with the following lines:

"We also observed two possible positions for RR binding in the TRPV5 SF, a higher one which seems to be preferred when the channel has not been opened coordinated at Asp542 and a lower one coordinated by Asn572 which is dominant when the channel has previously been

activated by PI(4,5)P₂. We propose that RR, in its attempt to traverse the pore like any other cation, encounters obstacles due to interaction with the Asn572 and its physicochemical properties, including charge, and size. These characteristics collectively impede its movement within the permeation pathway, causing blockage of the channel and consequent closure.”

10. Despite RR having a higher affinity for the channel, TRPV2-2APB+RR still appears to be in an open-like state, rather than inhibited or closed. It is worth investigating whether any other channel conformations were excluded during data processing, possibly representing an inactivated state. Additionally, why isn't PIP2 observed in the TRPV5 structure? It is possible that PIP2-containing particles were excluded from data processing because they represent only a subset of particles. The data processing flow chart shows many classes that could be refined to a decent resolution, making it interesting to explore these different conformations.

For the TRPV2_{2APB+RR} dataset, the dominant initial state was the C4 activated conformation presented in this paper. We have reported in the past that the addition of 2-APB induces significant heterogeneity in TRPV2 (Pumroy 2022 Nat Comm), and particularly produces a large percentage of particles in asymmetric states along the transition between the fully activated and fully inactivated conformations. The C4 inactivated conformation was the dominant stable conformation from that dataset, representing ~9% of particles with good TMD density, while the C4 activated conformation represented less than 1% of those particles. By comparison, ~33% of particles (66K out of 201K with good TMD density) sorted into the activated TRPV2_{2APB+RR} structure we present here. We did see evidence of the asymmetric intermediate states as we saw previously for the 2-APB only data, but they were still too heterogeneous to accurately build models, especially in the area around the RR binding site. As a result, we have softened the language we use to describe how RR affects the state dynamics of the 2-APB and RR treated dataset in the results and discussion section, but we have chosen to keep the structure currently presented in the paper and not add any additional structures.

For the TRPV5_{PIP2+RR} structure, we made an extensive effort to find classes in the dataset that had PIP2 density, but were not ultimately able to find any. Throughout the data processing phase, we did observe a significant number of particles containing RR within the pore where the N-terminal end of helix 6 and a segment at the outset of the TRP helix remained undefined. This region holds significance not only in terms of PI(4,5)P₂ binding but also because it undergoes conformational alterations during activation. Consequently, it was our conjecture that some of these particles could contain PIP2. Regrettably, our efforts to obtain a structure involving these particles proved unfruitful.

11. Possibly beyond the scope of this article, but I am very curious how your data set might benefit from 3D variability analysis and/or multi-body refinement for the two particle stacks individually and/or also combined? It would be great to see if undulations coincide with RR bindings, and/or improvements in resolutions using multi-body refinement approaches

We have performed 3D variability analysis on this data, but generally find that for our datasets it results in a continuum of movement rather than distinct classes. Particularly for TRPV2, that subtle movement is also present in the absence of RR so it seems to be an inherent property of the channel rather than a meaningful effect of RR binding.

In all but the TRPV2_{RR} dataset, RR binding is extremely robust and we do not observe classes/conformations without RR bound.

12. How the lipids are placed in the channel in these structures and compared to previously published structures? Do authors observe any differences in lipid movement or displacement in RR bound state? I think a figure is required for comparing the lipid presence or absence or displacement in different conformation of these channels.

We agree with the reviewer that the role of lipids is important when talking about TRP channels. We have already included some figures showing lipid density present in these maps (Appendix Figure S3 for the TRPV2 vanilloid lipid, Appendix Figure S9 for the TRPV5 PIP2 density), but we have added the TRPV2_{RR} vanilloid lipid density to Appendix Figure S3 for comparison.

For TRPV2, we observe some lingering lipid tail density at high contour levels in the vanilloid pocket, but it is much weaker than we see in the apo channel or in TRPV2_{RR}. This indicates that there is less lipid occupancy of this pocket, as we describe in the text. Unfortunately, the high degree of heterogeneity induced by 2-APB limited the quality of most other lipids bound to TRPV2.

We do not observe a significant change in lipid densities between TRPV5_{Apo} and TRPV5_{PIP2} apart from the added PIP2. The other lipids we observe in the TRPV5_{RR} and TRPV5_{PIP2+RR} are essentially identical to the lipids we observe in TRPV5_{Apo}.

Referee #3:

The manuscript entitled "Molecular Details of Ruthenium Red Pore Block in TRPV Channels" submitted by Pumroy et al and by Vera Moiseenkova-Bell lab is an interesting manuscript. Ruthenium Red has been used as a channel blocker for a long time. In this manuscript, the authors attempted to understand the binding site of the Ruthenium Red on TRPV2 and TRPV5, a thermosensitive and a non-thermosensitive ion channel respectively. Though this work is good and a lot of work has been included in this manuscript, at present this manuscript lacks some key aspects. My comments are not the quality of the data that has been provided. My comments are mainly to address the questions and confirming the findings.

We thank the reviewer for their thoughtful comments

Major concerns are:

1) No mutagenesis work has been done to conform the role of main residues that confer Ruthenium Red binding. Ideally these few key residues should have been mutated and electrophysiology data should be provided to show that efficacy of Ruthenium Red binding on the mutated channel. In absence of this mutagenesis followed by electrophysiology data

The reviewer may have missed the mutational experiments we performed, described in the Results section and shown in Supplementary Figures 13 and 15 (in our first submission), and now called Figure EV4 and Figure EV5, respectively, where we show a significant reduction in RR block upon mutation of key residues identified in our cryoEM structures.

2) Simulation data is required to supplement the binding of Ruthenium Red. The simulation data will provide more mechanistic details.

We agree with the reviewer that it would be ideal to include simulations of RR binding, Our primary constraint in performing these experiments lies in the lack of a suitable force field

for the RR ligand. While some investigations have successfully addressed the isolated or in complex examination of the ruthenium free ion, our situation is notably more intricate because RR is a composite molecule. Although the generation of these parameters from the ground up is not insurmountable, it necessitates an extensive and thorough independent study for their development and validation that is beyond the scope of this work.

3) Another major concern is the introduction section. The logic to pick up only TRPV2 and TRPV5 among all other TRP channels are not properly justified.

We explain in the introduction that we wanted to compare how two TRP channel selectivity filters with different properties could be inhibited by the same molecule. While there may be other TRP channels that would be interesting to add to this comparison, on a practical level it is not trivial to optimize the expression and purification of any new protein, let alone the notoriously tricky TRP channels.

4) Also use of 2APB as TRPV2 agonist is bit critical. 2APB is not a specific ligand for TRPV2 and it binds to many other ion channels. In that context also logic to use 2APB as a ligand for TRPV2 need justification. The point is there might be a different structure altogether if the ligand become highly-specific for a channel.

We agree with the reviewer that 2-APB is not a good ligand to use with TRPV2, but unfortunately there isn't really a better activating ligand for the channel as TRPV2 does not currently have robust and specific drugs. However, there is compelling evidence that 2-APB can actually act on some of these channels through unique and specific mechanisms, as we recently showed that the agreed upon 2-APB binding site in TRPV3 [Hu et al PNAS (PMID: 19164517), Singh et al NSMB (PMID: 30127359), Zubcevic et al eLife (PMID: 31070581), Deng et al NSMB (PMID: 32572252)] is not conserved in TRPV2 [Pumroy 2022 Nat Comm (PMID: 31566564)].

5) Is there any mutation known in the human disease data base/cancer data base that high-lights the key residues involved in Ruthenium Red binding on the TRPV2 and TRPV5? Especially in the pore region?

Although TRPV2 and TRPV5 have been heavily implicated in a variety of cancers and disease in recent years, in these cases the channels are either upregulated or downregulated rather than undergoing mutations to alter function. The cancer database cBioPortal reports some mutations close to the pore but none of them involve residues forming the RR binding site. In TRPV5, for example, the mutant M578V has been found in colorectal adenocarcinoma, M577I in skin cutaneous melanoma, or A576T and L569I in uterine corpus endometrial carcinoma, all of them located in the S6 helix.

Manuscript number: EMBOR-2023-58051V2

Title: Molecular Details of Ruthenium Red Pore Block in TRPV Channels

Author(s): Vera Moiseenkova-Bell, Tibor Rohacs, Andreas Leffler, Ruth Pumroy, José De Jesús-Pérez, Anna Protopopova, Julia Rocereta, Edwin Fluck, Tabea Fricke, and Bo Lee

Dear Prof. Moiseenkova-Bell

Thank you for your patience while we have editorially reviewed your revised manuscript. I am now writing with an 'accept in principle' decision, which means that I will be happy to accept your manuscript for publication once a few minor issues/corrections have been addressed, as follows.

- Your manuscript will be published as Report and therefore the Results and Discussion sections need to be combined.
- Please move the Data Availability section after Materials and Methods. Can you please add links that resolve directly to the datasets deposited in PDB?
- Please update the 'Conflict of interest' paragraph to our new 'Disclosure and competing interests statement'. For more information see <https://www.embopress.org/page/journal/14693178/authorguide#conflictsofinterest>
- Please remove the Author Contributions from the manuscript file and make sure that the author contributions in our online submission system are correct and up-to-date. The information you specified in the system will be automatically retrieved and typeset into the article. You can enter additional information in the free text box provided, if you wish.
- Please add a callout to Figure 2C in the text.
- Please add a title page to the Appendix with a table of content and page numbers.
- Source data .xls files: please sort them into separate folders per figure and then zip these folders together.
- Our production/data editors have asked you to clarify several points in the figure legends (see below). Please incorporate these changes in the manuscript and return the revised file with tracked changes with your final manuscript submission:
 - A) Please indicate the statistical test used for data analysis in the legend of figure EV 5d.
 - B) Please note that information related to n is missing in the legends of figures EV 1d; EV 4c, f.
 - C) Please note that the error bars are not defined in the legend of figure EV 4c.

If all remaining corrections have been attended to, you will then receive an official decision letter from the journal accepting your manuscript for publication in the next available issue of EMBO reports. This letter will also include details of the further steps you need to take for the prompt inclusion of your manuscript in our next available issue.

On a different note, I would like to alert you that EMBO Press offers a new format for a video-synopsis of work published with us, which essentially is a short, author-generated film explaining the core findings in hand drawings, and, as we believe, can be very useful to increase visibility of the work. This has proven to offer a nice opportunity for exposure i.p. for the first author(s) of the study.

Please see the following link for representative examples and their integration into the article web page:

<https://www.embopress.org/doi/full/10.15252/embj.2019103932>

Thank you for your contribution to EMBO reports.

Yours sincerely,

The authors addressed the editorial issues.

Prof. Vera Moiseenkova-Bell
University of Pennsylvania
Systems Pharmacology and Translational Therapeutics
3400 Civic Center Boulevard, Building 421
10-124 Smilow Center for Translational Research
Philadelphia, PA 19104-5158
United States

Dear Prof. Moiseenkova-Bell,

I am very pleased to accept your manuscript for publication in the next available issue of EMBO reports. Thank you for your contribution to our journal.

Yours sincerely,
